# Incremental Learning of Retrievable Skills For Efficient Continual Task Adaptation

**Daehee Lee**[♠,◇]**, Minjong Yoo**[♠]**, Woo Kyung Kim**[♠]**, Wonje Choi**[♠]**, Honguk Woo**[♠]

[♠]Sungkyunkwan University    [◇]Carnegie Mellon University
{dulgi7245, mjyoo2, kwk2696, wjchoi1995, hwoo}@skku.edu

## Abstract

Continual Imitation Learning (CiL) involves extracting and accumulating task knowledge from demonstrations across multiple stages and tasks to achieve a multi-task policy. With recent advancements in foundation models, there has been a growing interest in adapter-based CiL approaches, where adapters are established parameter-efficiently for tasks newly demonstrated. While these approaches isolate parameters for specific tasks and tend to mitigate catastrophic forgetting, they limit knowledge sharing among different demonstrations. We introduce IsCiL, an adapter-based CiL framework that addresses this limitation of knowledge sharing by incrementally learning shareable skills from different demonstrations, thus enabling sample-efficient task adaptation using the skills particularly in non-stationary CiL environments. In IsCiL, demonstrations are mapped into the state embedding space, where proper skills can be retrieved upon input states through prototype-based memory. These retrievable skills are incrementally learned on their corresponding adapters. Our CiL experiments with complex tasks in Franka-Kitchen and Meta-World demonstrate robust performance of IsCiL in both task adaptation and sample-efficiency. We also show a simple extension of IsCiL for task unlearning scenarios.

## 1 Introduction

Lifelong agents such as home robots are required to continually adapt to new tasks in sequential decision-making situations by leveraging knowledge from past experiences. However, many real-world domains pose substantial challenges for these lifelong agents; the complexity and ever-changing nature of these tasks make it difficult for agents to constantly adapt, leading to difficulties in retaining knowledge and maintaining operational efficiency [1]. For instance, a home robot agent, operating within a single household, needs to continuously adapt, learning specific tasks in various areas such as cooking assistance in the kitchen or cleaning in the bathroom. At the same time, it is crucial that the agent not only retains but also improves its proficiency in the tasks it has previously learned, ensuring that it maintains consistent efficiency throughout the home.

For these lifelong agents, Continual Imitation Learning (CiL) has been explored, in which an agent progressively learns a series of tasks by leveraging expert demonstrations over time to achieve a multi-task policy. Yet, CiL often encounters practical challenges: (1) the high costs and inefficiencies associated with comprehensive expert demonstrations [2] that are required for imitation, (2) frequently shifting tasks in dynamic, non-stationary environments, and (3) privacy concerns [3] related to learning from expert demonstrations. In this context, CiL faces significant issues in terms of cost, adaptability, and privacy, complicating its implementation in real-world scenarios.

---

Corresponding author: Honguk Woo (hwoo@skku.edu). Daehee Lee is currently a visiting scholar at Carnegie Mellon University.

38th Conference on Neural Information Processing Systems (NeurIPS 2024).

To address these challenges, our work focuses on incorporation of skill learning and fine-tuning in CiL, leveraging recent advancements in foundation models [4, 5]. These have been increased interests in continual task adaptation based on multiple adapters learned on a foundation model [6, 7]. The adapter-based learning approach allows for parameter isolation for individual tasks, thus enabling to mitigate catastrophic forgetting of previously learned knowledge in CiL. Motivated by this use of adapters, we develop IsCiL, a new adapter-based CiL framework that addresses the practical challenges of CiL aforementioned, by incrementally learning shareable skills from different demonstrations through multiple adapters. IsCiL facilitates sample-efficient task adaptation using the skills particularly in non-stationary CiL environments.

Specifically, in the IsCiL framework, a prototype-based skill incremental learning method is employed with a two-level hierarchy including smaller, more manageable adapters: skill retriever and skill decoder. The skill retriever is responsible for composing skills to complete given goal-reaching tasks. It utilizes skill prototypes, which are representative embeddings of skills, to retrieve the appropriate skill for input. The knowledge of each skill is contained within the adapter, which can modify its associated base model output. The skill decoder is responsible for producing short-horizon actions for state-skill pairs.

We evaluate IsCiL and several adapter-based continual learning baselines across scenario variations based on complex, long-horizon tasks in the Franka-Kitchen and Meta-World environments to assess sample efficiency, task adaptation, and privacy considerations. The baselines include adapter-based continual adaptation techniques as well as conventional continual imitation learning methods. Our results demonstrate that IsCiL achieves robust performance without requiring comprehensive expert demonstrations. This flexibility allows IsCiL to continually and efficiently adapt to varying sequences in different environments by leveraging any available expert data to learn useful skills, with tasks composed of diverse instructions and demonstrations.

In summary, the IsCiL framework enhances sample efficiency and task adaptation, effectively bridging the gap between adapter-based CiL approaches and the knowledge sharing across demonstrations. Comprehensive experiments demonstrate that IsCiL outperforms other adapter-based continual learning approaches in various CiL scenarios.

## 2   Related work

**Continual imitation learning.** To tackle the problem of catastrophic forgetting in continual learning, numerous studies have employed rehearsal techniques [8, 9, 10, 11], which involve replaying past experiences to maintain performance on previously learned tasks. Another approach involves utilizing additional model parameters to progressively extend the model architecture [12, 13, 14, 15, 16]. These methods adapt the model's structure over time to accommodate new tasks. However, rehearsal techniques exhibit high variability in forgetting depending on the replay ratio and often demand substantial training to incorporate new knowledge [17]. Progressive models, on the other hand, require stage identification during evaluation and often overlook unseen tasks [13]. In this work, we propose a CiL framework that enables effective learning and expansion without requiring rehearsal and stage identification, leveraging pre-trained goal-based model knowledge.

**Continual task adaptation with pre-trained models.** Several recent works use pre-trained models, accumulating knowledge continually through additional Parameter Efficient Tuning (PET) modules such as adapters [18, 17, 19, 20, 21, 6, 22]. These methods enhance the flexibility and scalability of continual learning systems. However, they suffer from inaccurate matching between adapter selection and trained knowledge, leading to a misalignment between the knowledge learned during training and the knowledge used during evaluation [17, 20], which hinders overall performance. In the realm of sequential decision making, some studies have explored adapting pre-trained models. In [6], the state space of tasks is fully partitioned, restricting its applicability in more integrated environments. Meanwhile, [7] relies on comprehensive demonstrations for learning, which may be impractical in real-world scenarios. Our study aims to enhance task adaptation efficiency by using incrementally generalized skills with accurate matching on state space.

**Skill adaptation.** Reinforcement learning research has enhanced fast adaptation through skill exploration [23] and skill priors [24], focusing on improving sample efficiency with offline datasets. Despite these advancements, adapting fixed skill decoders to new environments remains challenging. To overcome these limitations, skill-based few-shot imitation learning methods have been developed

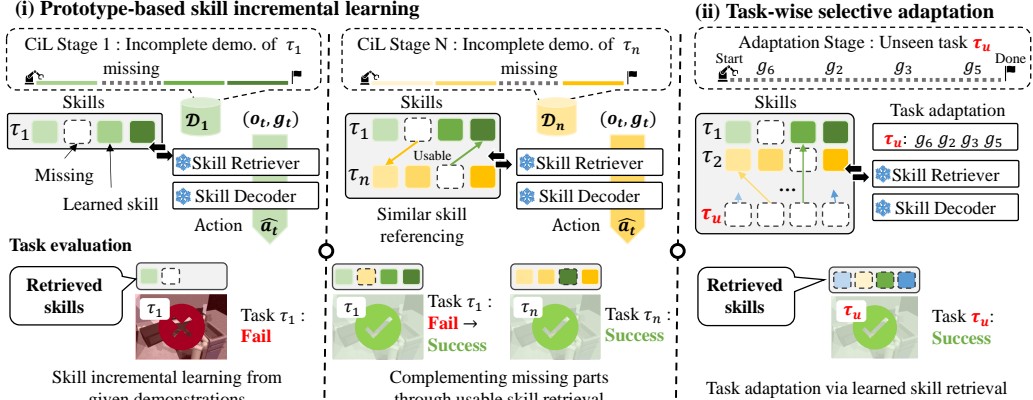

Figure 1: The scenario demonstrating how IsCiL enhances continual imitation learning efficiency through retrievable skills: (i) Prototype-based skill incremental learning: despite the failure of $\tau_1$, skills are incrementally learned from the available demonstrations. In later stages, missing skills for $\tau_1$ are retrieved from other tasks, achieving the resolution of $\tau_1$ and illustrating the reversibility and efficiency of retrievable skills. (ii) Task-wise selective adaptation: IsCiL effectively retrieves relevant learned skills, facilitating rapid task adaptation.

[25, 26]. However, these methods require extensive past data and struggle with scalability and generalization. Even skill-based approaches used in continual imitation learning [8] still require rehearsal data to mitigate knowledge loss and face difficulties addressing privacy issues through unlearning. Our IsCiL employs parameter-efficient skill adapters to prevent catastrophic forgetting and maintain efficiency, providing a scalable solution for unlearning.

## 3 Approaches

Our work addresses three key challenges of CiL: (1) data inefficiency, (2) non-stationarity, and (3) privacy concerns, by adopting retrievable skills in the CiL context. Specifically, our IsCiL framework not only enhances data-efficient continual task evaluation in a non-stationary environment but also supports unlearning as a task adaptation strategy, thereby mitigating privacy concerns.

### 3.1 Problem formulation

In CiL scenarios, we consider a data stream of task datasets $\{\mathcal{D}_i\}_{i=1}^p$, where $\mathcal{D}_i$ contains an expert demonstration $\mathcal{D}_i = \{d_i^1, ..., d_i^N\}$ for its associated task $\tau_i$. To effectively represent complex long-horizon tasks, each task $\tau_i$ is comprised of sub-goal list, $\tau = \{g_i^1, ..., g_i^M\}$. Each task dataset is sampled in a finite-horizon markov decision process $(\mathcal{S}, \mathcal{A}, \mathcal{P}, \mathcal{R}, \mu_0, H)$, where $\mathcal{S}$ is a state space, $\mathcal{A}$ is a action space, $\mathcal{P}$ is a transition probability, $\mathcal{R}$ is a reward function, $\mu_0$ is an initial state distribution, and $H$ is an environment horizon.

For demonstration $d = \{(s_t, a_t)\}_{t=1}^H$, a state $s_t \in S$ represents a tuple $(o_t, g_t)$ consisting of an observation $o_t$ and a sub-goal $g_t$. In our work, we represent sub-goals through language and use language-based goal embeddings for $g_t$ to achieve language-conditioned policies. Then, the objective of IsCiL is to obtain a multi-task policy $\pi^*$, by which the performance on the tasks in the data stream can be comparable to that of respective expert policies. This is formulated as

$$\pi^* = \underset{\pi}{\operatorname{argmin}} \left[ \mathbb{E}_i \left[ \sum_{\tau \in \mathcal{T}_i} \mathrm{KL}(\pi(\cdot|s) \| \tilde{\pi}_\tau(\cdot|s)) \right] \right] \tag{1}$$

where $\tilde{\pi}_\tau$ represents an expert policy for $\tau$ and $\mathcal{T}_i$ denotes a set of evaluation tasks at stage $i$. In this context, the evaluation tasks continuously vary across different stages.

## 3.2 Overall architecture

To effectively handle complicated CiL scenarios, we present the IsCiL framework which involves (i) **prototype-based skill incremental learning** and (ii) **task-wise selective adaptation**.

As illustrated in Figure 1, in (i) the prototype-based skill incremental learning, we use a two-level hierarchy structure with a skill retriever $\pi_R$ composing the skills for each sub-goal, and a skill decoder $\pi_D$ producing short-horizon actions based on state-skill pairs. For this two-level policy hierarchy, we employ a skill prototype-based approach, in which skill prototypes capture the sequential patterns of actions and associated environmental states, as observed from expert demonstrations. These prototypes serve as a reference for skills learned from a multi-stage data stream. Using these skill prototypes, we can effectively translate task-specific instructions or demonstrations into a series of appropriate skills.

Through this prototype-based skill retrieval method, the policy flexibly uses skills that are shareable among tasks, potentially learned in the past or future, for policy evaluation. This enables the CiL agent to effectively learn diverse tasks and rapidly adapt to variations, while incrementally accumulating skill knowledge from a multi-stage data stream. Furthermore, to facilitate sample-efficient learning and enhance stability in CiL, we employ parameter-efficient adapters that are continually fine-tuned on a base model. Each skill knowledge is encapsulated within a dedicated adapter and incorporated into the skill decoder $\pi_D$ to infer expert actions.

In (ii) the task-wise selective adaptation, we devise efficient task adaptation procedures in the policy hierarchy to adapt to specific tasks using incrementally learned skills. This enables the CiL agent to not only facilitate adaptation to shifts in task distribution (e.g., due to non-stationary environment conditions) but also support task unlearning upon explicit user request (e.g., due to privacy concerns).

Suppose that the smart home environment undergoes an upgrade with the installation of new smart lighting systems throughout the house. In this case, task-wise selective adaptation can be used for rapid adaptation by removing outdated control routines associated with the previous systems.

## 3.3 Prototype-based skill incremental learning

**State encoder and prototype-based skill retriever.** To facilitate skill retrieval from demonstrations, we encode observation and goal pairs $(o_t, g_t)$ into state embeddings $s_t$ using a function $f : (o_t, g_t) \mapsto s_t$. We implement $f$ as a fixed function to ensure consistent retrieval results for learning efficiency, mitigating the negative effects of input distribution shifts.

To effectively handle the multi-modality of the state distribution in non-stationary environments, we employ a skill retriever $\pi_R$. For this, we use multifaceted skill prototypes $\chi_z \in \mathcal{X}$, where $\mathcal{X}$ is the set of learned skill prototypes. These prototypes capture the sequential patterns of expert demonstrations associated with specific goal-reaching tasks.

$$\theta_z = \pi_R(s_t; \mathcal{X}) = h\left(\text{argmax}_{\chi_z \in \mathcal{X}} S(\chi_z, s_t)\right), \text{ where } S(\chi_z, s_t) = \max_{b \in \chi_z} \text{sim}(b, s_t) \quad (2)$$

Here, $h : \chi_z \mapsto \theta_z$ denotes a one-to-one function that maps each skill prototype $\chi_z$ to its dedicated adapter parameters $\theta_z$, while the similarity function $S$ is defined as the maximum similarity between state $s$ and bases $b \in \chi_z$. Each $\chi_z$ consists of multiple bases (e.g., 20 bases), and each basis $b$ is a representative vector containing its corresponding centroid, shaped identically to the state $s_t$.

**Adapter conditioned skill decoder.** To effectively use the knowledge of the pre-trained base model without forgetting, even in a non-stationary changing environment, the skill decoder is conditioned based on parameters. The skill decoder policy $\pi_D(\hat{a}_t|o_t, g_t; \theta_{\text{pre}}, \theta_z)$ operates with the skill adapter parameters $\theta_z$ and the pre-trained base model $\theta_{\text{pre}}$, using the Low-Rank Adaptation [27].

**Skill incremental learning.** To incrementally learn new retrievable skills, we update the skill prototype and adapter pair $(\chi_{z^*}, \theta_{z^*})$ for a novel skill $z^*$. The skill prototype $\chi_{z^*}$ is created by dividing a dataset of a single skill into several clusters based on similarity. From each cluster, a representative value is extracted to serve as the basis $b$, representing $z^*$. We use the KMeans algorithm [28] to determine these bases, ensuring that the number of bases $|\chi_z|$ adequately captures the diversity within the dataset of the novel skill. This multifaceted set of bases allows the skill prototype to capture an accurate multi-modal distribution of the skill represented in the state space, enabling effective retrieval as described in Eq. 2. In our experiment, $z^*$ is created for each sub-goal $g$ in the given dataset $\mathcal{D}_i$ for each stage $i$.

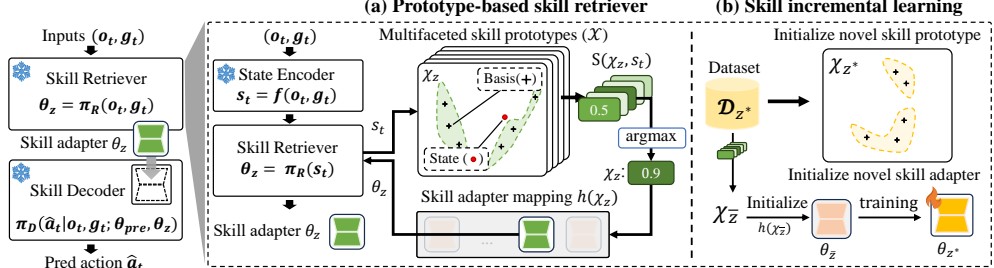

Figure 2: Overview of the IsCiL framework: (a) The prototype-based skill retriever sequentially utilizes a state encoder $f$, multifaceted skill prototypes $\mathcal{X}$, and a skill adapter mapping function $h$ to identify the skill adapter $\theta_z$. (b) Skill incremental learning involves the initialization and updating of the skill prototype $\chi_{z^*}$ and its corresponding adapter $\theta_{z^*}$.

The learning of the skill adapter is divided into two phases: initialization and update. During the initialization phase, $\theta_{z^*}$ is initialized using existing skill adapters. Predictions with the existing skill dataset and skill prototypes $\chi_z \in \mathcal{X}$ are used to identify the most frequently selected skill. Average scores are computed for each skill prototype using the dataset involved in training $z^*$, as defined in Eq. 2. The skill with the highest average score, denoted as $\bar{z}$, is selected. Consequently, $\theta_{z^*}$ is initialized by $\theta_{\bar{z}}$. Then, the initialized adapter is updated through the following imitation loss.

$$\mathcal{L}(o_t, g_t, a_t; \theta_z) = \|a - \pi_D(\hat{a}_t \mid o_t, g_t; \theta_{\text{pre}}, \theta_z)\|, \text{ where } \theta_z = \pi_R(o_t, g_t) \tag{3}$$

The novel skill $z^*$ is incorporated into the learned prototypes $\mathcal{X} \leftarrow \mathcal{X} \cup \chi_{z^*}$, and the novel prototype and adapter pair $(\chi_{z^*}, \theta_{z^*})$ updates the function $h$ for pair mapping. Figure 2 presents an overview of this methodology, along with the algorithm for incremental learning is detailed in Appendix B.1.

### 3.4 Task-wise selective adaptation

**Task evaluation.** Given the pre-trained model $\theta_{\text{pre}}$ and learned skill prototypes $\mathcal{X}$, for given inputs $(o_t, g_t)$ from the environment, IsCiL performs the following evaluation process.

$$\hat{a}_t \sim \pi_D(\hat{a}_t \mid o_t, g_t; \theta_{\text{pre}}, \theta_z), \text{ where } \theta_z = \pi_R(o_t, g_t; \mathcal{X}) \tag{4}$$

The evaluation process adapts to novel tasks and sub-goal sequences from the environment by modifying the goal $g_t$. This adjustment enables the inference of appropriate current actions, in a manner of similar to handling learned tasks. For example, a kitchen robot tailored to a specific user's kitchen setup can continuously and instantly adapt to changes in recipes without additional training.

**Task unlearning.** To ensure privacy protection for incrementally learned skills, our architecture allows for task unlearning by removing task-specific skill prototypes and adapters. In IsCiL, the separation of skill adapters for each task facilitates easy tagging of task information on each skill. When an unlearning request is given with a task identifier $\tau$, the corresponding skill prototypes and adapters are removed. This approach ensures exceptionally efficient and effective unlearning, aligning with the strong unlearning strategies in continual learning discussed in [3].

## 4 Experiments

### 4.1 Environments and data streams

To investigate the sample efficiency and adaptation performance, we construct complex CiL scenarios using diverse long-horizon tasks [29, 30, 31]. We then analyze the sample efficiency across different stages and tasks with three types of scenarios: *Complete*, *Semi*-complete, and *Incomplete*, depending on how the samples are utilized and shared. Each scenario consists of a pre-training stage followed by 20 CiL stages. Figure 3 illustrates these scenarios.

**Evolving Kitchen.** Evolving Kitchen is a data stream based on long-horizon tasks in the Franka-Kitchen environment [29, 30]. Each task requires sequentially achieving four out of seven sub-goals. The scenario consists of a pre-training stage in the environment with only four objects: kettle, bottom burner, top burner, and light switch, followed by continual adaptation to tasks involving seven objects.

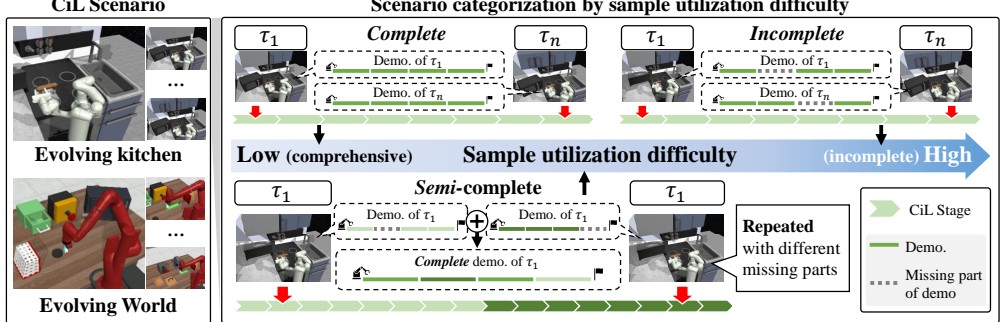

Figure 3: CiL scenarios including *Complete*, *Semi-Complete*, and *Incomplete*, categorized by sample utilization difficulty, based on the completeness of the demonstration for task performance: In *Complete*, each of the 20 CiL stages incrementally introduces new tasks featuring objects not encountered in the pre-training stage, along with full, comprehensive demonstrations for each task. In *Semi-Complete*, the first 10 stages are repeated twice, with tasks presented alongside incomplete demonstrations, where specific sub-goals are missing from the trajectories. In *Incomplete*, the same sequence of tasks from the *Complete* scenario is used, but all stages feature incomplete demonstrations, requiring the system to handle tasks with missing sub-goal trajectories.

**Evolving World.** Evolving World is a data stream based on the Meta-World environment [31] with long-horizon tasks, similar to [32, 33, 34]. Each task requires sequentially achieving four out of eight sub-goals. The scenario consists of a pre-training stage in the environment with only four objects, followed by continual adaptation to an entire environment with all eight Meta-World objects. More detailed configurations are provided in Appendix A.

## 4.2 Baselines and metrics

**Baselines.** We implement continual imitation learning and continual adaptation methods for sequential decision-making problems, which do not use rehearsal. First, we consider continual learning algorithms which involve full-model updates (**Seq**, **EWC** [35]). We also implement several continual adaptation approaches that utilize pre-trained models with adapters (**L2M** [6], **TAIL** [7]). L2M learns a key and adapter pair to modulate the pre-trained model, where the key is a retrievable state embedding similar to our prototypes. TAIL, unlike L2M, incrementally constructs task identifiers and corresponding adapters to modulate the pre-trained model with new task data without forgetting previous tasks. Each method is categorized based on the values used for adapter retrieval: a version that uses no additional identifiers, sub-goal identifiers (denoted as **-**$g$), and whole sub-goal sequences as single identifiers (denoted as **-**$\tau$). Additionally, we include a **Multi-task** learning approach as an oracle baseline, which retains all incoming data at each stage and utilizes it for training in subsequent stages. For all baselines, we use the same pre-trained goal-conditioned policy and a diffusion model [36, 37] as the base policy architecture. A detailed description of the baselines and their hyperparameters are provided in Appendix B.2.

**Metrics.** We use three metrics to report CiL performance: Forward Transfer (FWT), Backward Transfer (BWT), and Area Under Curve (AUC) [38, 7]. In our long-horizon tasks, these metrics rely on goal-conditioned success rates (GC), which measure the ratio of successfully completed sub-goals to the total sub-goals within each task [39].

- **FWT** (Forward Transfer): This evaluates the ability to learn tasks using previously learned knowledge. It is measured by the performance of a task when it occurs.
- **BWT** (Backward Transfer): This evaluates the impact of each learning stage on the performance of tasks learned in previous stages. It measures the change in task performance from past stages observed in the current stage.
- **AUC** (Area Under Curve): This represents the overall continual imitation learning performance in a scenario. It measures the average performance of tasks learned in the current stage over the remaining stages of the scenario.

For all metrics, **higher values indicate better** performance, with details provided in Appendix B.3.

Table 1: Overall performance on CiL scenarios of Evolving Kitchen and Evolving World: The rows represent baselines, categorized into sequential adaptation and adapter-based approaches, and oracle, respectively. The columns represent continual learning scenarios, where each scenario has 20 stages. Each scenario in the environment is categorized into *Complete*, *Semi*-complete, and *Incomplete*. The highest performance is highlighted in **bold** and the second highest performance is underlined.

| Stream | Evolving Kitchen-*Complete* | | | Evolving Kitchen-*Semi* | | | Evolving Kitchen-*Incomplete* | | |
|---|---|---|---|---|---|---|---|---|---|
| CiL-algorithm | FWT (%) | BWT (%) | AUC (%) | FWT (%) | BWT (%) | AUC (%) | FWT (%) | BWT (%) | AUC (%) |
| Pre-trained | - | - | $24.3_{\pm0.5}$ | - | - | $29.1_{\pm0.9}$ | - | - | $24.3_{\pm0.5}$ |
| Seq-FT | $90.9_{\pm2.6}$ | $-63.7_{\pm2.7}$ | $35.0_{\pm0.7}$ | $37.1_{\pm2.1}$ | $-25.1_{\pm2.7}$ | $16.5_{\pm0.7}$ | $32.7_{\pm4.3}$ | $-19.6_{\pm3.0}$ | $15.7_{\pm0.5}$ |
| EWC | $34.2_{\pm0.8}$ | $-19.5_{\pm4.2}$ | $17.1_{\pm2.7}$ | $27.2_{\pm1.3}$ | $-18.0_{\pm1.3}$ | $12.2_{\pm1.4}$ | $19.3_{\pm2.3}$ | $-3.2_{\pm11.3}$ | $10.4_{\pm1.7}$ |
| Seq-LoRA | $77.5_{\pm2.6}$ | $-55.2_{\pm1.8}$ | $28.3_{\pm1.5}$ | $37.4_{\pm3.8}$ | $-25.5_{\pm3.2}$ | $15.9_{\pm1.6}$ | $32.9_{\pm2.5}$ | $-19.9_{\pm2.9}$ | $14.5_{\pm0.2}$ |
| L2M | $24.7_{\pm4.8}$ | $-2.5_{\pm4.5}$ | $22.7_{\pm1.6}$ | $19.2_{\pm4.4}$ | $0.2_{\pm1.3}$ | $19.1_{\pm4.8}$ | $17.5_{\pm4.0}$ | $-2.0_{\pm3.2}$ | $15.8_{\pm4.8}$ |
| L2M-*g* | $38.2_{\pm3.4}$ | $-6.5_{\pm3.7}$ | $32.3_{\pm1.4}$ | $37.9_{\pm3.7}$ | $-4.5_{\pm3.1}$ | $32.1_{\pm1.2}$ | $37.5_{\pm10.0}$ | $-6.5_{\pm6.9}$ | $31.0_{\pm8.8}$ |
| TAIL-*g* | $85.3_{\pm8.0}$ | $-49.9_{\pm6.7}$ | $41.5_{\pm1.7}$ | $55.0_{\pm1.5}$ | $-21.1_{\pm2.2}$ | $37.2_{\pm2.4}$ | $53.2_{\pm1.7}$ | $-20.0_{\pm2.0}$ | $35.4_{\pm0.7}$ |
| TAIL-*τ* | $86.2_{\pm5.6}$ | $0.0_{\pm0.0}$ | $86.2_{\pm5.6}$ | $41.2_{\pm2.5}$ | $0.0_{\pm0.0}$ | $41.2_{\pm2.5}$ | $33.8_{\pm3.0}$ | $0.0_{\pm0.0}$ | $33.8_{\pm3.0}$ |
| IsCiL (ours) | $79.3_{\pm1.7}$ | $11.0_{\pm1.6}$ | $\mathbf{89.8_{\pm0.5}}$ | $68.1_{\pm2.2}$ | $8.6_{\pm0.6}$ | $\mathbf{75.8_{\pm1.8}}$ | $61.8_{\pm0.9}$ | $13.7_{\pm2.9}$ | $\mathbf{74.0_{\pm1.9}}$ |
| Multi-task | $93.3_{\pm1.7}$ | $-1.6_{\pm2.3}$ | $92.3_{\pm1.8}$ | $75.4_{\pm4.5}$ | $8.0_{\pm5.5}$ | $83.2_{\pm1.1}$ | $71.7_{\pm1.1}$ | $12.6_{\pm0.8}$ | $83.0_{\pm1.1}$ |
| **Stream** | Evolving World-*Complete* | | | Evolving World-*Semi* | | | Evolving World-*Incomplete* | | |
| CiL-algorithm | FWT (%) | BWT (%) | AUC (%) | FWT (%) | BWT (%) | AUC (%) | FWT (%) | BWT (%) | AUC (%) |
| Pre-trained | - | - | $0.0_{\pm0.0}$ | - | - | $0.0_{\pm0.0}$ | - | - | $0.0_{\pm0.0}$ |
| Seq-FT | $88.9_{\pm3.1}$ | $-73.6_{\pm4.2}$ | $24.9_{\pm0.4}$ | $38.9_{\pm5.9}$ | $-27.5_{\pm5.5}$ | $13.2_{\pm0.9}$ | $41.4_{\pm2.0}$ | $-33.0_{\pm2.0}$ | $12.2_{\pm0.8}$ |
| EWC | $25.7_{\pm3.8}$ | $-18.0_{\pm0.2}$ | $10.5_{\pm3.5}$ | $13.9_{\pm1.4}$ | $-9.1_{\pm1.8}$ | $6.2_{\pm1.8}$ | $18.2_{\pm2.8}$ | $-11.6_{\pm2.1}$ | $8.5_{\pm0.9}$ |
| Seq-LoRA | $85.6_{\pm2.9}$ | $-75.1_{\pm2.3}$ | $21.4_{\pm1.2}$ | $32.2_{\pm5.2}$ | $-18.2_{\pm4.9}$ | $16.0_{\pm2.3}$ | $38.1_{\pm1.6}$ | $-30.6_{\pm2.0}$ | $11.7_{\pm0.9}$ |
| L2M | $72.1_{\pm5.3}$ | $-6.6_{\pm2.1}$ | $65.9_{\pm3.3}$ | $41.0_{\pm2.1}$ | $6.3_{\pm3.0}$ | $47.0_{\pm0.7}$ | $26.1_{\pm1.1}$ | $5.7_{\pm2.8}$ | $31.4_{\pm2.0}$ |
| L2M-*g* | $64.2_{\pm3.9}$ | $-19.3_{\pm4.4}$ | $48.6_{\pm2.0}$ | $44.5_{\pm2.0}$ | $3.4_{\pm2.5}$ | $48.2_{\pm0.2}$ | $33.2_{\pm2.0}$ | $-0.6_{\pm0.9}$ | $33.1_{\pm2.2}$ |
| TAIL-*g* | $90.0_{\pm3.0}$ | $-56.8_{\pm0.4}$ | $39.5_{\pm2.9}$ | $43.2_{\pm7.8}$ | $-17.6_{\pm3.5}$ | $27.4_{\pm5.1}$ | $51.4_{\pm2.5}$ | $-21.4_{\pm0.6}$ | $32.5_{\pm2.3}$ |
| TAIL-*τ* | $85.7_{\pm5.9}$ | $0.0_{\pm0.0}$ | $\mathbf{85.7_{\pm5.9}}$ | $27.5_{\pm0.7}$ | $0.0_{\pm0.0}$ | $27.5_{\pm0.7}$ | $39.7_{\pm1.0}$ | $0.0_{\pm0.0}$ | $39.7_{\pm1.0}$ |
| IsCiL (ours) | $81.7_{\pm0.4}$ | $2.7_{\pm0.9}$ | $84.3_{\pm1.1}$ | $60.0_{\pm1.1}$ | $9.3_{\pm1.4}$ | $\mathbf{68.9_{\pm0.5}}$ | $63.2_{\pm1.5}$ | $8.7_{\pm2.7}$ | $\mathbf{71.2_{\pm4.2}}$ |
| Multi-task | $88.6_{\pm3.6}$ | $2.8_{\pm3.5}$ | $90.7_{\pm1.2}$ | $55.0_{\pm3.6}$ | $27.6_{\pm4.1}$ | $80.9_{\pm0.3}$ | $73.2_{\pm1.7}$ | $12.6_{\pm1.2}$ | $84.2_{\pm1.3}$ |

## 4.3 Overall performance : sample efficiency

Table 1 shows the CiL performance on Evolving Kitchen and Evolving World across three different scenarios (*Complete*, *Semi*, *Incomplete*). We compare the performance achieved by our framework IsCiL and other baselines (L2M, TAIL) with different conditioning values ($g$,$\tau$) for adapter retrieval. IsCiL consistently demonstrates superior performance in AUC across all scenarios, achieving between 84.5% and 97.2% of the oracle baseline (Multi-task learning). TAIL-$\tau$ shows the most competitive performance in the *Complete* CiL scenario across both environments. However, due to its isolated adapter for learning and evaluation, it fails to effectively utilize samples across stages.

L2M and L2M-$g$ exhibit relatively lower and less stable AUC in the Evolving Kitchen scenario. Conversely, in Evolving World-*Semi*, they surpass TAIL-$\tau$ in AUC. This demonstrates that they are capable of sharing different skills across stages. Despite this, they still struggle with accurately retrieving the correct skill or suffer from performance degradation due to knowledge overwriting. Unlike them, IsCiL effectively mitigates overwriting by maintaining distinct skill representations across stages. Both L2M-$g$ and TAIL-$g$, which aim to leverage sub-goal labels for CiL, struggle to maintain performance due to skill distribution shifts, leading to catastrophic forgetting of skills for sub-goals. These challenges reveal that relying solely on sub-goal labels may not be sufficient to sustain and share skills effectively across different stages and tasks.

Both Seq-FT and Seq-LoRA struggle with forgetting. This is evident in the *Complete* scenario, where Seq-FT achieves the highest FWT but shows the lowest BWT, leading to a decline in overall performance. EWC exhibits consistently lower performance, as the regularization used to preserve past knowledge significantly hinders learning on current tasks, leading to severe degradation in long-horizon tasks. Although EWC shows higher BWT compared to other sequential tuning baselines, its low FWT limits overall effectiveness.

Table 2: Task adaptation performance with unseen tasks: This is based on the existing Evolving World-*Complete* and Evolving Kitchen-*Complete*. In Evolving World, four novel tasks are introduced every four stages, while in Evolving Kitchen, two novel tasks are introduced every five stages. Metrics with the suffix -A denote performance based solely on adaptation tasks, while other metrics report performance across all tasks.

| Stream | Evolving Kitchen-*Complete* Unseen | | | | | Evolving World-*Complete* Unseen | | | | |
|---|---|---|---|---|---|---|---|---|---|---|
| Algorithm | FWT (%) | BWT (%) | AUC (%) | FWT-A (%) | AUC-A (%) | FWT (%) | BWT (%) | AUC (%) | FWT-A (%) | AUC-A (%) |
| Seq-FT | $72.3_{\pm1.6}$ | $-47.7_{\pm1.6}$ | $30.4_{\pm0.2}$ | $27.8_{\pm0.6}$ | $19.5_{\pm0.1}$ | $52.9_{\pm3.6}$ | $-26.7_{\pm1.8}$ | $30.1_{\pm2.1}$ | $16.3_{\pm1.8}$ | $24.0_{\pm2.6}$ |
| EWC | $21.0_{\pm15.9}$ | $-14.0_{\pm2.0}$ | $16.8_{\pm1.6}$ | $18.1_{\pm4.2}$ | $14.4_{\pm1.6}$ | $16.5_{\pm1.9}$ | $-8.1_{\pm0.8}$ | $9.6_{\pm2.6}$ | $6.1_{\pm1.3}$ | $8.3_{\pm2.1}$ |
| Seq-LoRA | $62.4_{\pm3.8}$ | $-41.5_{\pm3.3}$ | $25.4_{\pm0.9}$ | $28.1_{\pm0.0}$ | $18.2_{\pm0.0}$ | $45.2_{\pm0.4}$ | $-35.8_{\pm1.3}$ | $14.5_{\pm0.9}$ | $6.4_{\pm2.5}$ | $8.2_{\pm1.8}$ |
| L2M | $22.3_{\pm2.3}$ | $0.3_{\pm1.5}$ | $22.7_{\pm3.5}$ | $15.3_{\pm3.2}$ | $21.2_{\pm4.1}$ | $55.1_{\pm3.7}$ | $-1.4_{\pm3.3}$ | $53.6_{\pm1.0}$ | $40.3_{\pm2.4}$ | $41.2_{\pm2.0}$ |
| L2M-$g$ | $33.8_{\pm0.9}$ | $-4.3_{\pm1.2}$ | $30.0_{\pm0.4}$ | $22.2_{\pm0.6}$ | $24.1_{\pm0.7}$ | $43.3_{\pm1.6}$ | $-8.2_{\pm3.6}$ | $35.7_{\pm1.6}$ | $24.2_{\pm1.5}$ | $25.7_{\pm1.7}$ |
| TAIL-$g$ | $67.6_{\pm7.4}$ | $-34.9_{\pm5.4}$ | $36.8_{\pm3.2}$ | $34.7_{\pm2.2}$ | $30.1_{\pm1.0}$ | $53.2_{\pm1.4}$ | $-27.1_{\pm1.2}$ | $29.2_{\pm0.3}$ | $18.6_{\pm0.9}$ | $19.1_{\pm0.6}$ |
| IsCiL (ours) | $69.5_{\pm2.5}$ | $16.3_{\pm2.2}$ | $84.4_{\pm1.3}$ | $52.1_{\pm7.5}$ | $72.8_{\pm2.1}$ | $64.3_{\pm2.6}$ | $-0.5_{\pm3.5}$ | $63.9_{\pm0.6}$ | $45.8_{\pm4.7}$ | $45.3_{\pm0.9}$ |
| Multi-task | $85.3_{\pm1.7}$ | $3.7_{\pm1.8}$ | $88.8_{\pm0.0}$ | $70.8_{\pm0.0}$ | $79.0_{\pm0.1}$ | $85.4_{\pm0.9}$ | $5.6_{\pm0.5}$ | $90.4_{\pm0.5}$ | $78.3_{\pm2.9}$ | $85.9_{\pm0.4}$ |

Table 3: Overall performance with task unlearning as task adaptation: Additional stages for unlearning tasks that were learned during other stages are included for tests.

| Stream | Evolving Kitchen-*Complete* Unlearning | | | Evolving Kitchen-*Incomplete* Unlearning | | |
|---|---|---|---|---|---|---|
| Algorithm | FWT (%) | BWT (%) | AUC (%) | FWT (%) | BWT (%) | AUC (%) |
| TAIL-$\tau$ CLPU | $86.2_{\pm5.6}$ | $0.0_{\pm0.0}$ | $86.2_{\pm5.6}$ | $33.8_{\pm3.0}$ | $0.0_{\pm0.0}$ | $33.8_{\pm3.0}$ |
| IsCiL (ours) | $75.0_{\pm7.2}$ | $11.2_{\pm5.5}$ | $85.2_{\pm1.8}$ | $61.4_{\pm2.9}$ | $12.4_{\pm2.9}$ | $72.7_{\pm2.9}$ |

## 4.4 Task adaptation

Table 2 shows the unseen task adaptation ability of IsCiL, where only the sub-goal sequence of novel task is provided without demonstrations. This scenario extends the existing *Complete* CiL scenarios by periodically introducing novel tasks. Metrics labeled with the suffix -A indicate results from adaptation tasks, whereas the other metrics reflect performance on all tasks. For this scenario, we exclude TAIL-$\tau$ from comparison, as it lacks the ability to adapt to novel tasks.

IsCiL demonstrates superior task performance in both scenarios, which contributes to greater efficiency in task adaptation. Moreover, in Evolving Kitchen, IsCiL not only demonstrates task adaptation ability by achieving the highest FWT-A, but also significantly enhances its initial performance, raising FWT-A from 52.1 to an AUC-A of 72.8. TAIL-$g$ shows comparable performance in FWT for the Evolving Kitchen. However, it struggles with catastrophic forgetting, leading to a $-34.9$ negative BWT when faced with significant distribution shifts in sub-goal demonstrations. In Evolving World, L2M, which actively learns to share skills during training, outperforms TAIL-$g$. L2M is the only baseline achieving performance improvement on unseen tasks through CiL.

## 4.5 Task unlearning as adaptation

Table 3 measures CiL performance in scenario with task-level unlearning. For comparison, we use an adapter-based approach with parameter isolation-based continual learning private unlearning (CLPU) [3], extending TAIL to **TAIL-$\tau$ CLPU** and IsCiL without skill adapter initialization. Similar to IsCiL, CLPU learns tasks in isolated models tagged with specific task identifiers and handles unlearning requests by removing the corresponding model parameters of the target task. Both **TAIL-$\tau$ CLPU** and IsCiL ensure output distribution equality between the unlearned model and the model trained with the retained dataset. Thus, their CiL performance remains largely unaffected by unlearning.

Although IsCiL exhibits a slight performance degradation of $1.8\% \sim 5.2\%$ after unlearning, as reported in Table 1, it still demonstrates robustness by achieving a $115\%$ higher AUC compared to **TAIL-$\tau$ CLPU** in *incomplete* scenarios.

## 4.6 Analysis

**Rehearsal comparison.** Figure 4 compares the sample efficiency to retain learned knowledge between IsCiL and a rehearsal-based continual imitation learning approach, Experience Replay (ER) [40]. For ER, we adjust the number of stored samples per learning stage, while IsCiL does not store rehearsals

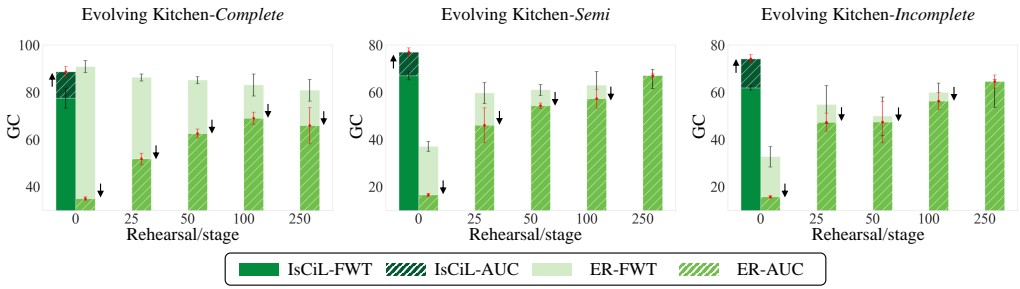

Figure 4: Comparison w.r.t. the number of rehearsals: The horizontal axis represents the amount of stored rehearsal data at each stage, while the vertical axis indicates goal-conditioned success rates (GC).

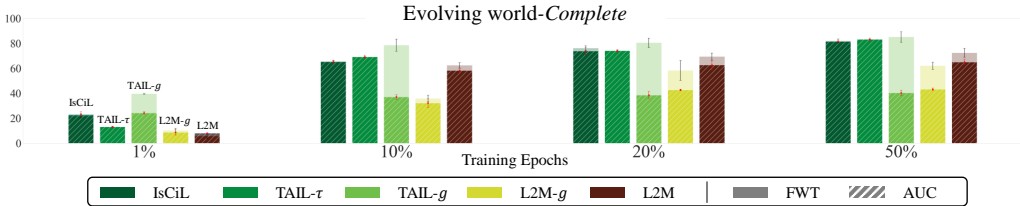

Figure 5: Comparison w.r.t. training resources: In all baselines, the plain bar graph represents FWT, while the bar graph with hatch marks represents AUC. The vertical axis indicates goal-conditioned success rates (GC).

for training. IsCiL achieves the highest AUC in all environments and is the only approach where AUC surpasses FWT. ER shows comparable FWT in *Complete*, but as the number of stored samples increases, FWT decreases, indicating that more rehearsals actually reduce training sample efficiency. In *Semi* and *Incomplete*, using 250 rehearsals (approximately 5% of the stage dataset) yields FWT comparable to IsCiL but rarely improves AUC.

**Limited training resource.** Figure 5 shows the computational efficiency of IsCiL in resource-constrained training settings, as discussed in [38]. In this experiment, the training resources is limited to 1% to 50% of those used in Table 1. IsCiL and TAIL-$\tau$ show robust performance for varied training resources. TAIL-$g$ shows higher FWT, as it trains the same sub-goal data on the same adapter, which excels in learning new tasks, but it fails to retain that knowledge. However, using skill data from different stages to update the same adapter makes it vulnerable to skill distribution shifts in CiL; this ends up with significant AUC degradation.

## 4.7 Ablation

Table 4 investigates the impact of the number of prototype bases on CiL performance, showing that increasing the number of bases improves both AUC and result stability, particularly around K=10. Results are reported based on units ($g$ and $\tau$) used to construct new skill prototypes and the corresponding number of bases. IsCiL with a single base fails to effectively learn task knowledge, achieving similar performance to L2M in Table 1, due to insufficient representation of the skill distribution. Additionally, the IsCiL framework maintains positive BWT scores,

Table 4: Ablation on IsCiL skill prototype

| Stream | Evolving kitchen-*Complete* | | |
|---|---|---|---|
| Ablations | FWT (%) | BWT (%) | AUC (%) |
| IsCiL $g, |\chi_z| = 20$ | $79.3_{\pm 1.7}$ | $11.0_{\pm 1.6}$ | $89.8_{\pm 0.5}$ |
| IsCiL $g, |\chi_z| = 1$ | $28.9_{\pm 10.2}$ | $2.3_{\pm 5.7}$ | $30.6_{\pm 12.2}$ |
| IsCiL $g, |\chi_z| = 5$ | $63.1_{\pm 4.0}$ | $2.7_{\pm 7.3}$ | $66.5_{\pm 7.9}$ |
| IsCiL $g, |\chi_z| = 10$ | $76.4_{\pm 6.5}$ | $8.2_{\pm 4.0}$ | $83.9_{\pm 2.7}$ |
| IsCiL $g, |\chi_z| = 25$ | $77.1_{\pm 1.8}$ | $11.9_{\pm 1.7}$ | $88.2_{\pm 1.1}$ |
| IsCiL $g, |\chi_z| = 50$ | $81.5_{\pm 2.8}$ | $7.9_{\pm 4.9}$ | $89.4_{\pm 1.2}$ |
| IsCiL $\tau, |\chi_z| = 20$ | $57.8_{\pm 16.6}$ | $10.9_{\pm 1.8}$ | $67.2_{\pm 17.2}$ |
| IsCiL $\tau, |\chi_z| = 80$ | $84.3_{\pm 6.7}$ | $5.0_{\pm 7.5}$ | $89.5_{\pm 8.3}$ |

demonstrating its ability to leverage future samples to enhance past performance. IsCiL with $\tau$, which constructs new skills based on entire task trajectories, required more bases in proportion to the increase in the number of transitions involved in constructing the skill trajectory to maintain stability.

## 5   Conclusion

In this study, we presented the IsCiL framework to address key challenges in continual imitation learning (CiL). Our approach incorporates adapter-based skill learning, leveraging multifaceted skill prototypes and an adapter pool to effectively capture the distribution of skills for continual task adaptation. IsCiL specifies enhanced sample efficiency and robust task adaptation, effectively bridging the gap between adapter-based CiL approaches and the need for knowledge sharing across staged demonstrations. Comprehensive experiments demonstrate that IsCiL consistently outperforms other adapter-based continual learning approaches in various CiL scenarios.

**Limitations.** Like other adapter-based CiL approaches, IsCiL requires extra computation for evaluation, which can create overhead, especially in resource-constrained environments. It also depends on sub-goal sequences for training and evaluation, adding complexity and resource demands. Another limitation is determining the appropriate size of the adapter parameters, which depends on the performance of the pre-trained base model and the degree of task shift, making optimal adaptation challenging. Moreover, balancing the stability of the embedding function with the prototype size remains an area that requires further refinement to achieve optimal performance.

## Acknowledgement

This work was supported by Institute of Information & communications Technology Planning & Evaluation (IITP) grant funded by the Korea government (MSIT) RS-2022-II220043 (2022-0-00043), Adaptive Personality for Intelligent Agents, RS-2022-II221045 (2022-0-01045), Self-directed multi-modal Intelligence for solving unknown, open domain problems, RS-2019-II190421, Artificial Intelligence Graduate School Program (Sungkyunkwan University), the National Research Foundation of Korea (NRF) grant funded by the Korea government (MSIT) (No. RS-2023-00213118) and by Samsung Electronics.

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

# A Environment and Data Stream Details

## A.1 Franka Kitchen

We conduct experiments on the Franka kitchen environment [29, 30]. Each Franka kitchen task comprises of 4 sub-goals, from total pool of 7: microwave, kettle, bottom burner, top burner, light switch, slide cabinet, and hinge cabinet. Observation is a 60-dimensional vector, which is a combination of the positions and velocities of 7-DoF robot arm and interacting objects. We express sub-goal information using language embedding. The target sub-goal of the current state is acquired by a pre-defined environmental reward and task, and a sub-goal sequence to solve. We use 24 tasks in the 'mixed' dataset from the D4RL [29]. In the pre-training stage, we train the model only on tasks comprised of following four sub-goals: kettle, bottom burner, top burner, light switch.

| Pre-training stage | Continual imitation learning stages |
|---|---|

Figure 6: Example of a multi-stage Meta-World environment in our continual imitation learning scenarios.

## A.2 Multi-stage Meta World

We conduct experiments on the multi-stage variation of the Meta-World environment [31, 34]. Each Meta-World task comprises of 4 sub-goals from total pool of 8: puck, box, handle, drawer, lever, button, door, and stick. The environments are divided into different scenarios based on which 4 out of 8 objects are placed on the table. For each environment, tasks are defined according to the sequence in which the 4 sub-goals must be achieved. Observation is a 140-dimensional vector, which contains the positions and velocities of the 4-DoF robot arm and all interacting objects in the environment. In this environment, sub-goal information is also expressed using language embedding. The expert dataset is collected using a heuristic expert policy provided by Meta-World [31]. In the pre-training stage of Meta World, we train the model on 24 tasks on a environment consisting of four objects: a puck, a drawer, a button, and a door.

## A.3 Data Stream

**Evolving Kitchen** Tables 5 and 6 display the detailed configurations of the Evolving Kitchen in our CiL scenario. Each task involves sequentially solving its respective sub-goals. The underlined sub-goals (e.g., kettle) are those missing in the *Semi Complete* and *Incomplete* scenarios.

**Evolving World** Tables 7 and 8 display the detailed configurations of our Evolving World CiL scenario. Similarly, the Evolving World is also presented in the same way as the Evolving Kitchen.

**Unseen Task Adaptation** Tables 9 and 10 show the detailed configurations of unseen tasks for our Evolving Kitchen-*Complete* Unseen and Evolving World-*Complete* Unseen. In the Evolving Kitchen, 2 new tasks appear every 5 stages. In the Evolving World, 4 new tasks appear every 4 stages. Each new task includes only the sequence of sub-goals that must be completed in order, without any demonstrations.

Table 5: Evolving Kitchen-*Complete & Incomplete* data stream task configuration

**Evolving Kitchen-*Complete & Incomplete***

| Task | Sub-goal 1 | Sub-goal 2 | Sub-goal 3 | Sub-goal 4 |
|---|---|---|---|---|
| $\tau_1$ | microwave | kettle | top burner | light switch |
| $\tau_2$ | kettle | bottom burner | top burner | slide cabinet |
| $\tau_3$ | microwave | bottom burner | top burner | slide cabinet |
| $\tau_4$ | kettle | bottom burner | light switch | slide cabinet |
| $\tau_5$ | microwave | kettle | light switch | slide cabinet |
| $\tau_6$ | kettle | bottom burner | top burner | hinge cabinet |
| $\tau_7$ | microwave | kettle | top burner | hinge cabinet |
| $\tau_8$ | microwave | kettle | slide cabinet | hinge cabinet |
| $\tau_9$ | kettle | light switch | slide cabinet | hinge cabinet |
| $\tau_{10}$ | microwave | kettle | bottom burner | hinge cabinet |
| $\tau_{11}$ | kettle | bottom burner | slide cabinet | hinge cabinet |
| $\tau_{12}$ | kettle | bottom burner | light switch | hinge cabinet |
| $\tau_{13}$ | microwave | top burner | light switch | hinge cabinet |
| $\tau_{14}$ | microwave | kettle | bottom burner | slide cabinet |
| $\tau_{15}$ | microwave | kettle | light switch | hinge cabinet |
| $\tau_{16}$ | microwave | bottom burner | top burner | light switch |
| $\tau_{17}$ | kettle | top burner | light switch | slide cabinet |
| $\tau_{18}$ | microwave | bottom burner | top burner | hinge cabinet |
| $\tau_{19}$ | microwave | bottom burner | slide cabinet | hinge cabinet |
| $\tau_{20}$ | microwave | bottom burner | light switch | slide cabinet |

Table 6: Evolving Kitchen-*Semi* data stream task configuration

**Evolving Kitchen-*Semi***

| Task | Sub-goal 1 | Sub-goal 2 | Sub-goal 3 | Sub-goal 4 |
|---|---|---|---|---|
| $\tau_1$ | microwave | kettle | top burner | light switch |
| $\tau_2$ | kettle | bottom burner | top burner | slide cabinet |
| $\tau_3$ | microwave | bottom burner | top burner | slide cabinet |
| $\tau_4$ | kettle | bottom burner | light switch | slide cabinet |
| $\tau_5$ | microwave | kettle | light switch | slide cabinet |
| $\tau_6$ | kettle | bottom burner | top burner | hinge cabinet |
| $\tau_7$ | microwave | kettle | top burner | hinge cabinet |
| $\tau_8$ | microwave | kettle | slide cabinet | hinge cabinet |
| $\tau_9$ | kettle | light switch | slide cabinet | hinge cabinet |
| $\tau_{10}$ | microwave | kettle | bottom burner | hinge cabinet |
| $\tau_{11}$ | microwave | kettle | top burner | light switch |
| $\tau_{12}$ | kettle | bottom burner | top burner | slide cabinet |
| $\tau_{13}$ | microwave | bottom burner | top burner | slide cabinet |
| $\tau_{14}$ | kettle | bottom burner | light switch | slide cabinet |
| $\tau_{15}$ | microwave | kettle | light switch | slide cabinet |
| $\tau_{16}$ | kettle | bottom burner | top burner | hinge cabinet |
| $\tau_{17}$ | microwave | kettle | top burner | hinge cabinet |
| $\tau_{18}$ | microwave | kettle | slide cabinet | hinge cabinet |
| $\tau_{19}$ | kettle | light switch | slide cabinet | hinge cabinet |
| $\tau_{20}$ | microwave | kettle | bottom burner | hinge cabinet |

Table 7: Evolving World-*Complete & Incomplete* data stream task configuration

**Evolving World-*Complete & Incomplete***

| Task | Sub-goal 1 | Sub-goal 2 | Sub-goal 3 | Sub-goal 4 |
|---|---|---|---|---|
| $\tau_1$ | door | handle | button | box |
| $\tau_2$ | puck | drawer | stick | lever |
| $\tau_3$ | handle | puck | lever | door |
| $\tau_4$ | button | drawer | box | stick |
| $\tau_5$ | door | handle | box | button |
| $\tau_6$ | lever | stick | drawer | puck |
| $\tau_7$ | lever | puck | handle | door |
| $\tau_8$ | stick | button | drawer | box |
| $\tau_9$ | handle | button | box | door |
| $\tau_{10}$ | drawer | stick | lever | puck |
| $\tau_{11}$ | puck | lever | door | handle |
| $\tau_{12}$ | stick | button | box | drawer |
| $\tau_{13}$ | handle | button | door | box |
| $\tau_{14}$ | drawer | lever | stick | puck |
| $\tau_{15}$ | puck | lever | handle | door |
| $\tau_{16}$ | stick | box | button | drawer |
| $\tau_{17}$ | handle | door | box | button |
| $\tau_{18}$ | stick | drawer | puck | lever |
| $\tau_{19}$ | door | puck | lever | handle |
| $\tau_{20}$ | box | drawer | button | stick |

Table 8: Evolving World-*Semi* data stream task configuration

**Evolving World-*Semi***

| Task | Sub-goal 1 | Sub-goal 2 | Sub-goal 3 | Sub-goal 4 |
|---|---|---|---|---|
| $\tau_1$ | door | handle | button | box |
| $\tau_2$ | puck | drawer | stick | lever |
| $\tau_3$ | handle | puck | lever | door |
| $\tau_4$ | button | drawer | box | stick |
| $\tau_5$ | door | handle | box | button |
| $\tau_6$ | lever | stick | drawer | puck |
| $\tau_7$ | lever | puck | handle | door |
| $\tau_8$ | stick | button | drawer | box |
| $\tau_9$ | handle | button | box | door |
| $\tau_{10}$ | drawer | stick | lever | puck |
| $\tau_{11}$ | door | handle | button | box |
| $\tau_{12}$ | puck | drawer | stick | lever |
| $\tau_{13}$ | handle | puck | lever | door |
| $\tau_{14}$ | button | drawer | box | stick |
| $\tau_{15}$ | door | handle | box | button |
| $\tau_{16}$ | lever | stick | drawer | puck |
| $\tau_{17}$ | lever | puck | handle | door |
| $\tau_{18}$ | stick | button | drawer | box |
| $\tau_{19}$ | handle | button | box | door |
| $\tau_{20}$ | drawer | stick | lever | puck |

**Unlearning Scenario** In the Unlearning Scenario, 1 learned task is unlearned every 5 learning stages. In the Evolving Kitchen Unlearning scenario, $\tau_4$, $\tau_8$, $\tau_{13}$, and $\tau_{17}$ are sequentially unlearned.

# B  Experiment Details

## B.1  IsCiL Implementation

IsCiL consists of two modules: a skill retriever, $\pi_R$, and a skill decoder, $\pi_D$. The skill retriever $\pi_R$ includes three components: a state encoder $f$, skill prototypes $\mathcal{X}$, and a skill adapter mapping function $h$. Each skill prototype $\chi_z$ in $\mathcal{X}$ is composed of 20 bases $b$. To modulate skill decoder $\pi_D$, we use Low Rank Adaptation(LoRA) [27]. In our experiment, we used 4-rank LoRA adapters for skill adapter. IsCiL training and evaluation process follows :

Table 9: Evolving Kitchen-*Complete* Unseen task configuration

| Evolving World-*Complete* Unseen | | | | |
|---|---|---|---|---|
| **Stage** | **Sub-goal 1** | **Sub-goal 2** | **Sub-goal 3** | **Sub-goal 4** |
| 5 | microwave | kettle | top burner | slide cabinet |
| 5 | microwave | kettle | top burner | top burner |
| 10 | microwave | kettle | top burner | light switch |
| 10 | kettle | bottom burner | top burner | light switch |
| 15 | kettle | top burner | slide cabinet | hinge cabinet |
| 15 | microwave | top burner | slide cabinet | hinge cabinet |
| 20 | microwave | top burner | light switch | slide cabinet |
| 20 | kettle | top burner | light switch | hinge cabinet |

Table 10: Evolving World-*Complete* Unseen task configuration

| Evolving World-*Complete* Unseen | | | | |
|---|---|---|---|---|
| **Stage** | **Sub-goal 1** | **Sub-goal 2** | **Sub-goal 3** | **Sub-goal 4** |
| 4 | door | handle | button | box |
| 4 | puck | drawer | stick | lever |
| 4 | handle | puck | lever | door |
| 4 | button | drawer | box | stick |
| 8 | door | handle | box | button |
| 8 | puck | lever | drawer | stick |
| 8 | handle | lever | puck | door |
| 8 | box | drawer | stick | button |
| 12 | box | handle | door | button |
| 12 | lever | drawer | stick | puck |
| 12 | handle | lever | puck | door |
| 12 | box | drawer | stick | button |
| 16 | door | handle | box | button |
| 16 | puck | drawer | stick | lever |
| 16 | handle | puck | lever | door |
| 16 | box | drawer | stick | button |
| 20 | door | handle | box | button |
| 20 | lever | drawer | stick | puck |
| 20 | door | handle | lever | puck |
| 20 | box | drawer | stick | button |

---

**Algorithm 1** IsCiL Skill Incremental Learning

---

1: State encoding function $f$, Skill retriever $\pi_R$
2: Skill decoder $\pi_D$, Pre-trained parameter $\theta_{\text{pre}}$
3: Skill adapter mapping function $h$
4: **for** each stage $i$ in CiL Stages **do**
5:     **for** each sub-goal $g$ in task dataset $D_i$ **do**
6:         $D_i^g \leftarrow \{(o, g') \in D_i \mid g' = g\}$ // filter transitions related to the current sub-goal $g$
7:         $S_i^g \leftarrow \{f(o_t, g_t) \mid (o_t, g_t) \in D_i^g\}$ // encode states from the filtered dataset into state embeddings
8:         $\mathcal{X}^g \leftarrow \{\text{argmax}_{\chi_z \in \mathcal{X}} S(\chi_z, s_t) \mid s_t \in S_i^g\}$ // retrieve the most relevant skill prototypes for each state $s_t$
9:         $\chi_{\bar{z}} \leftarrow \text{Mode}(\mathcal{X}^g)$ // select the most frequently retrieved skill prototype from the set
10:         $\theta_{z^*} \leftarrow h(\chi_{\bar{z}})$ // map the selected skill prototype $\chi_{\bar{z}}$ to its skill adapter via $h$
11:         Update $\theta_{z^*}$ using Eq. (3) // update the skill adapter based on task-specific learning
12:         $\mathcal{X} \leftarrow \mathcal{X} \cup \chi_{z^*}$ // append the new skill prototype to the skill set for future retrieval
13:         Update the mapping function $h$ to map $\chi_{z^*}$ to the updated adapter $\theta_{z^*}$ // update $h$ with the new skill adapter
14:     **end for**
15: **end for**

---

**Algorithm 2** IsCiL Evaluation

---

1: State encoding function $f$, Skill retriever $\pi_R$
2: Skill decoder $\pi_D$, Pre-trained parameter $\theta_{\text{pre}}$
3: **while** not done **do**
4:     $s_t = f(o_t, g_t)$ // encode state
5:     $\theta_z = \pi_R(s_t)$ // retrieve skill
6:     $\hat{a}_t \sim \pi_D(o_t, g_t; \theta_{\text{pre}}, \theta_z)$ // decode the skill
7: **end while**

---

### B.2 Baselines

**Seq-FT & Seq-LoRA** Sequential Fine Tuning(Seq-FT) is a method that updates the entire model sequentially. The variation, Seq-LoRA, is used to determine how effectively the fixed pre-trained model can utilize its knowledge. Due to poor performance at very low ranks, Seq-LoRA was trained with a 64-rank adapter in our environment.

**EWC**[41] Elastic Weight Consolidation (EWC) regularizes the weight update by using the Fisher information matrix for each network parameter. For our experiment, we adopted the online version of EWC, which updates the Fisher information at each stage by exponential moving average, following the methods in [38, 7].We use the hyperparameter alpha, set to 10, to determine the regularization strength. For updating the online Fisher information matrix $\bar{F}_i$, we use the Fisher information matrix calculated at the current stage and apply the following formula for regularization: $\bar{F}_i = \gamma F_{i-1} + (1 - \gamma)F_i$, where $\gamma$ is set to 0.9.

**L2M**[6] L2M is an adapter-based continual learning method consisting of keys and their corresponding adapters. When an input is provided, L2M uses a similarity function to search for the key corresponding to that input. Input is converted to a query and utilized to search for a key, where key is a vector with the same shape as the query. Each key is then updated to maximize its similarity to the data point associated with it. Finally, the data is used to update the adapter that corresponds to the key value found through that data. This method maximizes the diversity of key usage frequency by adjusting the similarity between keys and input values during the training phase for learning new tasks [18, 6]. In our implementation, we use the normalized state value as the input query to find the key in L2M. For L2M-$g$, we use the normalized embedding of the state concatenated with the given conditioned sub-goal information directly as a query. Our adapter pool consists of 100 adapters, each being a 4-rank LoRA adapter.

**TAIL**[7] TAIL directly assigns an adapter to the given task using the task's identifier. We directly map the given identifier to the corresponding adapter. TAIL-$g$ uses a 4-rank adapter, while TAIL-$\tau$ uses a 16-rank adapter.

**Multi-task** At each stage, the model learns from the given data and stores all data for the next stage. The data stored in the buffer is mixed with the data from each stage in a 1:1 ratio for training the model.

**ER**[40] Experience Replay (ER), similarly, retains knowledge by storing a subset of the current stage's data for the next stage. The data stored in the buffer is mixed with the data from each stage in a 1:1 ratio for training the model.

**CLPU**[3] Continual Learning Private Unlearning (CLPU) [3] is a method for managing continual learning and unlearning. In a continual learning scenario, tasks that require maintenance are trained using existing models, while data that may require unlearning is trained on independent model parameters, tagged with when and through which task each model was trained. When an unlearning request for a specific task or training stage is received, the corresponding model parameters are completely removed to eliminate the influence of the target unlearning task from the model. CLPU provides highly efficient and powerful unlearning performance with a single delete operation for tasks learned in a continual learning scenario. In our experiment, we integrate the unlearning approach CLPU with TAIL-$\tau$, which conducts training through task information-based searches, to handle unlearning requests in continual imitation learning as a comparative method.

### B.3 Metric

We report 3 metrics for CiL performance for tasks: Forward Transfer(FWT), Backward Transfer(BWT), Area Under Curve(AUC) [38, 7]. In multi-stage environment task, we report performance using the goal-conditioned success rates (GC), which evaluate the average success rate of successfully completed sub-goals out of $N$ sub-goals in the task.

- **FWT**: $\text{FWT}_\tau = \frac{1}{|I_\tau|} \sum_{i \in I_\tau} C_{\tau,i}$ where $\tau$ is task and $C_{\tau,i}$ represents the GC score of task $\tau$ at stage $i$. and $I_\tau$ is set of stage indices where task $\tau$ is trained in the CiL scenario.

- **BWT**: $\text{BWT}_\tau = \frac{1}{|I_\tau|} \sum_{i \in I_\tau} \left( \frac{1}{p-i-1} \sum_{j=i+1}^{p} (C_{\tau,j} - C_{\tau,i}) \right)$, where $p$ is the final stage at which task $\tau$ is available. In the case where $p = i$ BWT is 0.

- **AUC**: $\text{AUC}_\tau = \frac{1}{|I_\tau|} \sum_{i \in I_\tau} \left( \frac{1}{p-i} \sum_{j=i}^{p} C_{\tau,j} \right)$, represent the the overall performance of continual learning, internally including FWT and BWT. In the case where $p = i$, $\text{AUC}_\tau$ is $\text{FWT}_\tau$.

The final reported metric is the average across all tasks $\tau \in \mathcal{T}$. For all metrics, higher values indicate better performance.

## B.4 Scenario training details

**Pre-trained Base Model and Stage Settings** Table 11 shows the hyperparameters and the architecture of the model we used as the base model for all baselines. Table 12 shows the common hyperparameters used to train the model for each stage in our experiments.

Table 11: Pre-trained model configure

| Hyperparameter | Value |
|---|---|
| Diffusion Model | DDPM [36] |
| Denoising step | 128 |
| Schedule | Linear |
| Linear start | 1e-4 |
| Linear end | 2e-2 |
| Block | MLP |
| The number of layers | 6 |
| hidden dimension | 512 |
| Layer normalization | yes |

Table 12: Continual imitation learning default hyperparameters

| Hyperparameter | Value |
|---|---|
| Learning rate | 5e-4 |
| Optimizer | Adam |
| Epochs/stage | 5000 |

**Pre-trained model performance** Table 13 shows the learning performance of the pre-trained model and its adaptation performance for tasks learned in scenarios without any prior training. In Evolving World, where new objects are added and the environment changes significantly, the pre-trained model failed to successfully complete any sub-goals of tasks.

Table 13: Pre-trained model performance

| **Stream**(Total phase) | Evolving Kitchen base | Evolving World-*Complete* | Evolving World-*Semi* | Evolving World-*Incomplete* |
|---|---|---|---|---|
| Evolving Kitchen Pre-trained | $98.8_{\pm 0.0}$ | $24.3_{\pm 0.5}$ | $29.1_{\pm 0.9}$ | $24.3_{\pm 0.5}$ |

| **Stream**(Total phase) | Evolving World Base | Evolving Kitchen-*Complete* | Evolving Kitchen-*Semi* | Evolving Kitchen-*Incomplete* |
|---|---|---|---|---|
| Evolving World Pre-trained | $100.0_{\pm 0.0}$ | $0.0_{\pm 0.0}$ | $0.0_{\pm 0.0}$ | $0.0_{\pm 0.0}$ |

## B.5 Compute Resources

**Computing machine** Our experimental platform is powered by an AMD 5975wx CPU and 2x RTX 4090 GPUs. The operating system used is Ubuntu 22.04.4 LTS, with Nvidia driver version 535.171.04 and CUDA version 12.2.

**Software Detail** We utilized jax 0.4.24, jaxlib 0.4.19, and flax 0.8.2 for our implementation.

**Training time** In the context of the Evolving Kitchen, each scenario involves training with three different seeds. The training duration averages 2 minutes per stage, with each stage consisting of 5000 epochs. Each scenario comprises 20 such stages, culminating in a total training time of 2 hours for a single experiment.

# C Additional Experiments

## C.1 Main experiment extension

**Training curve**. Figure 7 shows training curves of Evolving-kitchen *Complete* and *Incomplete* on Table 1. The curves provide a clear illustration of the performance progression of IsCiL and baseline methods, making changes in key metrics over the course of training easily observable.

## C.2 Analysis

**Skill adapter rank**. Table 14 shows the results of the ablation study on the performance of CiL based on the rank of the skill adapter. Overall, the 1-rank adapter in Evolving Kitchen demonstrates sufficient, or even superior, adaptation performance. However, in Evolving World, the 1-rank adapter leads to lower overall performance, indicating that some skills cannot be fully learned with a 1-rank adapter, resulting in a decline in performance.

**Skill decoder pre-trained model quality**. Table 15 shows the results of the ablation study on performance changes based on the quality of the pre-trained model (skill decoder). The quality of the pre-trained model varies with the number of objects included in the tasks used to pre-train the model. A decrease in the quality of the pre-trained model leads to a performance drop in both TAIL-$\tau$ and IsCiL, as the number of objects is reduced from 4 to 1.

**Scenario task sequence variation**. Table 16 shows the results of the task sequence variation analysis. We report the average performance for four different task sequences in Evolving Kitchen-*Complete*. The performance of all tasks at the final stage is not significantly affected. Since TAIL-$\tau$ learns independently for each task ID, there was no performance change with different sequences, and IsCiL also showed similar performance, indicating that task sequence variation had minimal impact on overall outcomes.

**Computational efficiency**. In our framework, skill retrieval and adaptation occur at each time step. Despite this continuous process, the impact on inference time and computational demands is minimal. Through our implementation on JAX, we observed that factors like compile optimization had a more significant effect on performance than model size. As a result, IsCiL demonstrates fast evaluation times, with retrieval and adaptation processes taking 3.6ms and 3.0ms, respectively, which ensures that IsCiL remains highly efficient during inference.

Additionally, the memory overhead required for the adapted model is minimal, with the skill adaptation adding only 0.37% to 1.48% additional parameters compared to the pre-trained model, depending on the LoRA rank (1 to 4). For skill retrieval, the parameter size of each skill prototype is relatively small, accounting for approximately 0.3% of the total model size. Furthermore, the inclusion of adapters in the skill decoder only increases the FLOPs by 3.13% of the pre-trained model, demonstrating that the retrieval and adaptation processes are computationally efficient and have a negligible impact on resource consumption.

Table 14: Ablation study on the skill adapter rank in Evolving Kitchen-*Complete* and Evolving World-*Complete*.

| Stream | | Evolving Kitchen-*Complete* | | | Evolving World-*Complete* | | |
|---|---|---|---|---|---|---|---|
| Rank | CiL-algorithm | FWT (%) | BWT (%) | AUC (%) | FWT (%) | BWT (%) | AUC (%) |
| | L2M-$g$ | $30.2_{\pm2.1}$ | $2.6_{\pm1.0}$ | $33.0_{\pm1.6}$ | $56.8_{\pm3.5}$ | $-16.9_{\pm5.2}$ | $41.6_{\pm1.3}$ |
| 1 | TAIL-$g$ | $93.2_{\pm2.5}$ | $-54.3_{\pm1.6}$ | $45.7_{\pm1.3}$ | $77.0_{\pm5.0}$ | $-47.9_{\pm1.9}$ | $34.6_{\pm3.7}$ |
| | IsCiL | $89.2_{\pm4.0}$ | $2.7_{\pm3.0}$ | $\mathbf{91.6_{\pm1.8}}$ | $73.6_{\pm5.1}$ | $-3.3_{\pm3.9}$ | $\mathbf{70.9_{\pm3.3}}$ |
| | L2M-$g$ | $38.2_{\pm3.4}$ | $-6.5_{\pm3.7}$ | $32.3_{\pm1.4}$ | $64.2_{\pm3.9}$ | $-19.3_{\pm4.4}$ | $48.6_{\pm2.0}$ |
| 4 | TAIL-$g$ | $85.3_{\pm8.0}$ | $-49.9_{\pm6.7}$ | $41.5_{\pm1.7}$ | $90.0_{\pm3.0}$ | $-56.8_{\pm0.4}$ | $39.5_{\pm2.9}$ |
| | IsCiL | $79.3_{\pm1.7}$ | $11.0_{\pm1.6}$ | $\mathbf{89.8_{\pm0.5}}$ | $81.7_{\pm0.4}$ | $2.7_{\pm0.9}$ | $\mathbf{84.3_{\pm1.1}}$ |

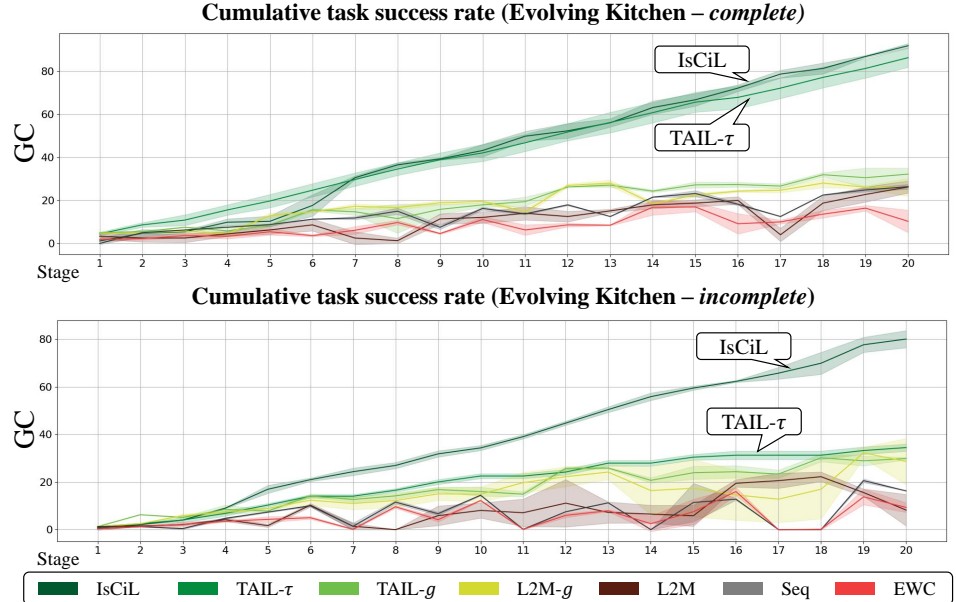

Figure 7: Evolving Kitchen-*complete* and Evolving Kitchen-*incomplete* training curves represent the cumulative task success rate up to a given stage. The goal conditioned success rate(GC) is scaled such that achieving success in all tasks by the final stage is represented as 100%. This result corresponds to the data presented in Table 1.

Table 15: Ablation study on the quality of the skill decoder pre-trained model in Evolving Kitchen-*Complete* and *Incomplete*.

| Stream | | Evolving Kitchen-*Complete* | | | Evolving Kitchen-*Incomplete* | | |
|---|---|---|---|---|---|---|---|
| CiL-algorithm | Pre-training | FWT (%) | BWT (%) | AUC (%) | FWT (%) | BWT (%) | AUC (%) |
| TAIL-$\tau$ | 1 object | $72.8_{\pm7.9}$ | $0.0_{\pm0.0}$ | $72.8_{\pm7.9}$ | $28.8_{\pm0.7}$ | $0.0_{\pm0.0}$ | $28.8_{\pm0.7}$ |
| | 2 object | $87.2_{\pm4.6}$ | $0.0_{\pm0.0}$ | $87.2_{\pm4.6}$ | $35.9_{\pm2.6}$ | $0.0_{\pm0.0}$ | $35.9_{\pm2.6}$ |
| | 4 object | $86.2_{\pm5.6}$ | $0.0_{\pm0.0}$ | $86.2_{\pm5.6}$ | $33.8_{\pm3.0}$ | $0.0_{\pm0.0}$ | $33.8_{\pm3.0}$ |
| IsCiL | 1 object | $60.0_{\pm4.0}$ | $2.1_{\pm4.2}$ | $62.1_{\pm0.8}$ | $42.1_{\pm7.3}$ | $5.4_{\pm3.2}$ | $47.0_{\pm4.6}$ |
| | 2 object | $78.9_{\pm5.1}$ | $6.4_{\pm3.7}$ | $84.9_{\pm1.3}$ | $56.7_{\pm3.2}$ | $12.0_{\pm2.3}$ | $67.3_{\pm1.6}$ |
| | 4 object | $79.3_{\pm1.7}$ | $11.0_{\pm1.6}$ | $89.8_{\pm0.5}$ | $61.8_{\pm0.9}$ | $13.7_{\pm2.9}$ | $74.0_{\pm1.9}$ |

Table 16: Analysis of task sequence variation in the CiL scenario of Evolving Kitchen-*Complete*.

| Stream | | Evolving Kitchen-*Complete* | | |
|---|---|---|---|---|
| CiL-algorithm | Task sequence | FWT (%) | BWT (%) | AUC (%) |
| IsCiL | Seq. 1 | $79.3_{\pm1.7}$ | $11.0_{\pm1.6}$ | $89.8_{\pm0.5}$ |
| | Seq. 2 | $80.4_{\pm2.9}$ | $4.4_{\pm2.7}$ | $87.6_{\pm1.9}$ |
| | Seq. 3 | $63.5_{\pm3.1}$ | $13.0_{\pm3.2}$ | $76.0_{\pm4.8}$ |
| | Seq. 4 | $89.6_{\pm1.9}$ | $1.3_{\pm1.2}$ | $90.8_{\pm0.9}$ |
| IsCiL | Average | $78.2_{\pm10.0}$ | $7.4_{\pm5.3}$ | $86.1_{\pm6.6}$ |
| TAIL-$\tau$ | Average | $86.2_{\pm5.6}$ | $0.0_{\pm0.0}$ | $86.2_{\pm5.6}$ |

### C.3 Scalability

**LIBERO**. Figure 8 provides a comprehensive visualization of the Skill Retriever in the LIBERO-goal scenario. The visualization highlights how the retriever successfully identifies and shares skills across different stages of CiL. This demonstrates its adaptability in handling varied states and tasks, showing its potential effectiveness even in complex LIBERO environments.

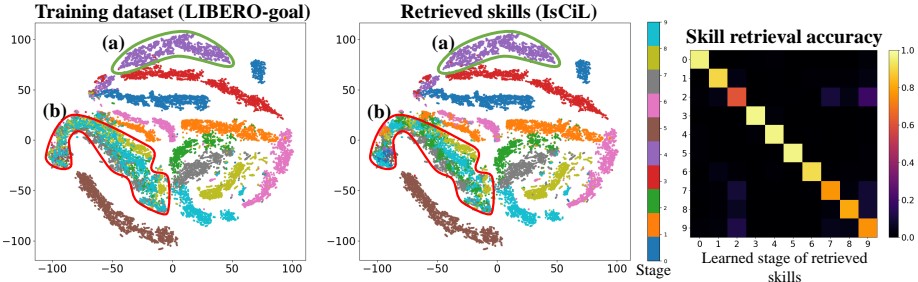

Figure 8: **Visualization of Skill Retriever on the LIBERO-goal Scenario. Left**: T-SNE visualization of the state space of the existing dataset for each stage. **Middle**: Visualization of the stages where skills retrieved by the Skill Retriever, after all CiL stages of learning. **Right**: Map showing the stages of each dataset and the retrieved skills. This demonstrates that the Skill Retriever can find skills capable of handling the given state, even in the LIBERO scenario. Additionally, in task-specific parts (a), it accurately retrieves the skills, and in parts showing similar behaviors (b), it shares skills.

