# OpenReview forum: "Incremental Learning of Retrievable Skills For Efficient Continual Task Adaptation"
_NeurIPS.cc/2024/Conference — NeurIPS 2024 poster_

### Official Review · Reviewer_DhgS · 2024-07-03

**Soundness:** 3
**Presentation:** 2
**Contribution:** 2
**Rating:** 5
**Confidence:** 3

**Summary:**

This paper tackles the problem of continual imitation learning, where an agent needs to continually adapt to new tasks through imitation learning. The paper proposed IsCiL, a method that utilizes prototype-based skill incremental learning to gradually grow a repository of skill prototypes that can be retrieved and adapted to new tasks. Evaluated on Franka-kitchen and meta world, IsCil showed superior performance over previous CiL methods.

**Strengths:**

The problem of continual imitation learning is very important. The overall idea of utilizing skills to achieve continual imitation learning is promising. The experimental results of IsCiL are well-presented, and thoroughly conducted.

**Weaknesses:**

Overall the contribution seems incremental. The idea of prototype-based skill learning, parameter-efficient adaptation via LoRA, as well as the idea of tackling lifelong learning through skills have already been explored by previous works, and it is not clear what the key innovation of this work is.

Some of the important concepts in this paper, such as skill prototype, base, and skill adaptor, are not properly defined, making it hard to comprehend how they are generated and used. For example, how exactly are the pre-trained base model and the adaptor weights combined? Where does the pre-trained base model come from? How are the skill prototypes initilized?

This paper assumes access to a set of subgoals. In practice, this seems like a strong assumption, as these subgoals essentially segment the demonstration trajectories and implicitly define a set of short-horizon skills. Considering that these sub-goals are also used for generating and selecting skill prototypes, it raises the concern that these sub-goal specifications are doing the heavy lifting.

**Questions:**

Although not necessary, I’d be interested to see the performance of IsCiL on more challenging benchmarks such as LIBERO. Also, given that the metrics measured by this work are intuitively visible from the training curve, it would be great to show some training curves of IsCiL and baseline methods.

Minor questions:
- How many skills are updated for each demonstration transition? It seems that only the retrieved skill is updated for each transition. Wouldn’t this be an inefficient use of data?
- How exactly does IsCiL determine when to add new skills? Do we need a human to manually identify new skills?

**Limitations:**

The limitations is adequately addressed.

---

> ### Author Rebuttal · Authors · 2024-08-06
>
> Thank you for your thorough and insightful review of our paper. Here, we respond to your comments and address the issues.
> > W1. The idea of tackling lifelong learning through skills have already been explored by previous works, and it is not clear what the key innovation of this work is.
>
> Our contribution lies in proposing a novel retrieval-based framework for realistic CiL (rehearsal-free, incomplete demonstration), demonstrating its skill-sharing ability with positive backward transfer and efficient knowledge management via unlearning. IsCiL is a distinct study with different problems, situations, and objectives compared to recent lifelong learning work using skills[8].
>
> Here are the differences between the latest skill-based lifelong imitation learning research and our study in terms of problem context and objectives:
> - **Existing research**[8]:
>    * Uses rehearsal
>    * Uses comprehensive demonstrations
>    * Learned skills are mutable
>    * Skills are shared by a high-level model and consolidation
>    * Avoids catastrophic forgetting
> - **IsCiL**:
>    * Rehearsal-free
>    * Uses comprehensive/incomplete demonstrations
>    * Learned skills are immutable
>    * Skills are shared through retrieval
>    * Bidirectional Skill Transfer(sample efficiency/skill sharing)
>    * Supports Task unlearning
>
> We demonstrated through the IsCiL framework and the experiments in Section 4 that the performance of skill retrieval and the consistency of the skill adapter are crucial for CiL performance in various situations. Therefore, both studies are important as they address different key aspects of lifelong learning.
> > W2. Provide more details of framework components.
>
> For clarity, we provide a concise overview of the overall structure of IsCiL(Section 3.3).
> - Skill Retriever: Returns the skill adapter capable of performing the given observation and subgoal(state).
>     - State encoder: Encode the given input into state.
>     - Multi-faceted prototype: Parameters representing skill precisely as bases that can be searched through the neierest neighbor by given state.
>     - Skill adapter: Adapter that can be added to the skill decoder, implemented through LoRA.
> - Skill Decoder: The model that directly infers actions from the given input.
>
> Actions of Each Component Based on the Scenario follows:
> Pre-training (Stage 0 or Given)
> - Pre-training: The base skill decoder is trained, and its parameters remain unchanged during the CiL stages.
>
> CiL Scenario (Stages 1-20)
> - Training (Skill incremental earning)
>     - Skill prototype initialization: Prototypes of skill obtained through centroids of KMeans.
>     - Skill adapter training: Adapters are trained using the transitions of the given skill.
>     - Append skill set: The trained skill prototype and adapter pair are added to the skill set and remain immutable in subsequent stages.
> - Evaluation
>     - Skill Retrieval: retrieve skill via given state
>     - Skill Decoding: The selected skill’s adapter combined with pre-trained model to infer actions.
>
> > W3. The assumption of access to sub-goals seems strong and raises the concern that these sub-goal specifications are doing the heavy lifting.
>
> The subgoal assumption in rehearsal-free CiL does not directly solve the forgetting problem or significantly enhance performance. Moreover, in our work, the sequence of subgoals represents distinct tasks, which is common in long-horizon multi-task settings.
> Here are the reasons:
> - Even within the same subgoal, the distribution of skills varies at each stage. We confirm this through the performance of the Tail-g and L2M-g baselines, which use subgoal labels directly. TAIL-g trains the adapter for each subgoal but experiences overall performance degradation due to skill distribution shift and forgetting. L2M-g also shows even lower FWT than TAIL-g, as making both the adapter and retriever learnable each time makes it more unstable.
> - Long-horizon task planning often uses language-based subgoal labels. All the environments we used involve long-horizon tasks that require the sequential execution of multiple sub-tasks (Section 4.1). Including shared sub-task goals in the data helps identify the task being performed[10].
>
> Therefore, subgoal specification is not unique to our methodology. Instead, it is a general setting that underscores the importance of knowledge sharing through our skill retriever.
> > Q1. It would be great to show some training curves of IsCiL and baseline methods, as well as performance on more challenging benchmarks like LIBERO.
>
> We added the learning curve from Table 1 and the applicability of IsCiL to the LIBERO benchmark in the *global rebuttal PDF*.
> > Q2. How many skills are updated for each demonstration transition? It seems that only the retrieved skill is updated for each transition. Wouldn’t this be an inefficient use of data?
>
> In our experiments, IsCiL creates 3 to 4 skills per stage, corresponding to the sub-goals in the demonstration. Updating only the skill adapter associated with a given transition in IsCiL offers several advantages:
> - It is robust to forgetting, which is critical for long-horizon tasks where a single mistake can lead to a significant performance drop. In CiL without rehearsal, if a new skill transition updates an already learned old skill, there is a risk of forgetting the updated old skill.
> - It allows for stable storage and removal of skill. Since skills are shared through retrieval, having adapters with precise knowledge related to the retrieved prototype is advantageous for skill evaluation and management.
> - It is training cost-efficient. If a single transition were to update multiple skills, the training cost would increase significantly.
>
> Therefore, IsCiL is effective in CiL scenarios without rehearsal and with incomplete expert demonstrations.
> > Q3. How exactly does IsCiL determine when to add new skills?
>
> New skills are appended at each stage without requiring manual decisions from the user.

---

> > ### Comment · Reviewer_DhgS · 2024-08-09
> >
> > I thank the authors for the clarifications and the additional experiments, which strengthened the paper. I've therefore raised my score.

---

> > > ### Author Response · Authors · 2024-08-09
> > > **Thank you!**
> > >
> > > Thank you for raising your score from 4 to 5. We truly appreciate your consideration and constructive feedback. We will incorporate the clarifications and discussion into the final version.
> > >
> > > The suggested training curve allowed us to intuitively demonstrate the performance of IsCiL, which is highly valuable. Additionally, the discussion about sub-goals helped us clarify the contributions of IsCiL, and we are pleased that this has further strengthened our work.

---

### Official Review · Reviewer_kNto · 2024-07-11

**Soundness:** 3
**Presentation:** 3
**Contribution:** 3
**Rating:** 6
**Confidence:** 3

**Summary:**

The paper presents IsCiL, an approach to continual learning that addresses the limitations of knowledge sharing in traditional Continual Imitation Learning methods. IsCiL uses a prototype-based skill incremental method where each skill is represented by prototype embeddings and skill adapter parameters for LoRA adaptation. The state encoder encodes observations and subgoals into state embeddings, while the skill retriever matches these to existing skill prototypes during inference. The method is evaluated in environments like Franka-Kitchen and Meta-World, demonstrating its ability to learn and adapt without needing complete expert demonstrations. The results show IsCiL's performance in task adaptation and its ability to perform task unlearning for privacy concerns.

**Strengths:**

The paper introduces a novel prototype-based skill retrieval mechanism that effectively learns skill prototypes and adapters. The proposed method is evaluated in environments like Franka-Kitchen and Meta-World, showing improvements over TAIL when knowledge sharing across demonstrations is important (semi and incomplete settings). While skill adapters and prototypes themselves are not particularly novel, the authors' contribution lies in the innovative method of continual learning through on-the-fly adapter retrieval. The authors additionally show that the method can enable skill unlearning without significant degradation in performance. The paper is well-structured, with clear explanations of the state encoder, skill retriever, and skill decoder components.

**Weaknesses:**

1. The retrieval and adaptation processes at every time step might lead to increased inference time and resources, which is not thoroughly analyzed in the paper.
2. Handling overlapping skills appears to not be handled or need manual intervention to unlearn, which is not ideal for maintaining performance across an increasing number of tasks.
3. The paper lacks a detailed analysis of the computational overheads and scalability issues associated with maintaining a prototype-based memory and multiple adapters, which is important for real-world applications.
4. A major assumption in the paper is the high costs and inefficiencies associated with comprehensive expert demonstrations. It is not well motivated why obtaining a single comprehensive demonstration would be more cost-effective than multiple incomplete demonstrations. This should be further motivated.

**Questions:**

1. Does retrieval and adaptation occur at every time step? How does this affect inference time and compute demands?
2. The method seems to require manual selection of the task identifier for task unlearning. How does the method handle conflicting skills that need to override or combine with existing prototypes? How are skills consolidated as the number and complexity of skills learns grows?
3. Can the unlearning process be applied at different granularities (e.g., specific sub-tasks or stages within a task), or is it only applicable at the (sub-)task level?

**Limitations:**

The authors do address limitations but could include more transparency on the inference time and resource use, which the paper does not thoroughly analyze. Secondly, handling a large amount of skill prototypes without additional consolidation.

---

> ### Author Rebuttal · Authors · 2024-08-06
>
> Thank you for your thorough and insightful review of our paper. We greatly appreciate your constructive feedback and for highlighting our contributions! Here, we respond to your comments and address the issues.
> > W1, Q1. The retrieval and adaptation processes at every time step might lead to increased inference time and resources. Does retrieval and adaptation occur at every time step? How does this affect inference time and compute demands?
>
> The retrieval and adaptation processes occur at each step and have a negligible impact on inference time. The IsCiL evaluation times for the pre-trained model (retrieval and adaptation) are 3.6ms and 3.0ms, respectively, making IsCiL even faster. This is because, in our implementation on JAX, factors such as compile optimization had a greater impact than the model size.
>
> The memory required for the adapted model during inference demands only an additional 0.37\% to 1.48\% parameters compared to the pre-trained model, depending on the LoRA rank (1 to 4). For skill retrieval, each skill requires parameters amounting to (572; total state dimension) * (20; bases size), which is about 0.3\% of our model size.
>
> The computation of adding adapters to the skill decoder increases the FLOPs by 3.13\% of the pre-trained model.
>
> > W2, Q2. How does the method handle conflicting skills and consolidate them as the number and complexity of learned skills grow?
>
> IsCiL incrementally accumulates skills by saving the adapter and prototype pair of each skill for every Continual Imitation Learning (CiL) stage, continuously accumulating overlapping skills as well(Section 3.3). Additionally, learned skills remain immutable after they are initialized and learned until an unlearning request is made.
>
> The reasons for IsCiL adopting append-only approach are as follows:
> - Even with multiple overlapping skills, the skill retriever can find the appropriate skill through a nearest neighbor search of the encoded state.
> - Continuously appended skills make it easier and simpler to manage privacy issues immediately. While consolidating skill adapters could improve scalability, this approach might require more complex processes or additional storage space for backups when removing specific knowledge [1].
>
> In general, as the number of skills increases, the complexity of the search also increases linearly. Skill retrieval can achieve optimized search times through GPU parallel processing and dense vector-based similarity search optimization methods [7].
> > W3. The paper lacks a detailed analysis of the computational overheads and scalability issues associated with maintaining a prototype-based memory and multiple adapters.
>
> We analyze these issues by examining the performance changes of IsCiL based on the scalability of prototypes and adapters. Prototype scalability results are reported in Table 4, and adapter scalability according to rank is presented in *global rebuttal Exp 1*.
> > W4. A major assumption in the paper is the high costs and inefficiencies associated with comprehensive expert demonstrations. Why obtaining a single comprehensive demonstration would be more cost-effective than multiple incomplete demonstrations?
>
> Our CiL scenarios assume long-horizon and complex tasks, making it challenging to collect comprehensive demonstrations that require executing multiple actions flawlessly. However, obtaining incomplete demonstrations, where each skill is present to perform parts of these long-horizon tasks, is much easier and not limited by the length of the demonstration. This is similar to the difference between filming a one-take movie and recording short YouTube videos.
>
> Table 1’s TAIL-$\tau$ exemplifies the issue that other CiL methodologies require comprehensive expertise, which is expensive in the long-horizon environments commonly used in real-world applications. It is difficult to combine task knowledge without forgetting when dealing with incomplete demonstrations. IsCiL ensures that even if a task initially fails, subsequent stages can reliably accumulate and share skills through retrieval, improving task success rates without additional processes. This advantage allows for easy editing and compensation for unsuccessful parts of tasks, while minimizing concerns about forgetting other skills in real-world scenarios.
>
> > Q3. Can the unlearning process be applied at different granularities (e.g., specific sub-tasks or stages within a task), or is it only applicable at the (sub-)task level?
>
> In Section 3.4 and Table 3, we demonstrate the unlearning capability using tagged metadata(task ID by sub-goal sequence) on the skill adapter. Similar to CLPU, using an isolated nature for the target unlearning adapter(model) in a continual learning scenario provides the advantage of enabling direct and immediate unlearning through metadata tagging.
>
> **IsCiL can be applied at different granularities**. This can be achieved combining a straightforward yet practical approach of adding more metadata tags (e.g., task information, learned stage, security level) to adapters all together. This allows for finding adapters related to user requests (e.g., specific stages within a task) and using them for unlearning. Although there may be a search overhead, the unlearning process itself remains fast and minimally impacts other knowledge.
>
> Furthermore, **unlearning is also possible when the target is a skill trajectory**(in the same format used for learning) rather than tagged metadata. IsCiL allows for skill retrieval through multifaceted prototypes, enabling the retrieval of specific skills. The same method used for adapter initialization in skill incremental learning in Section 3.3 can be applied and extended for this purpose. Thus, to remove a specific skill, providing the skill trajectory allows for its removal even without tagged metadata.

---

> > ### Comment · Reviewer_kNto · 2024-08-10
> > **Thank you for the comments**
> >
> > Thank you for your comments and for providing the additional analysis and clarifications on efficiency and scalability. As a result, I have increased my score.

---

> > > ### Author Response · Authors · 2024-08-11
> > > **Thank you!**
> > >
> > > Thank you for raising your score from 5 to 6. We truly appreciate your consideration and constructive feedback. We will, of course, incorporate this analysis into the final version.
> > >
> > > The discussion on how to effectively search for and consolidate the numerous learned skills will be a valuable future direction for IsCiL.

---

### Official Review · Reviewer_B1dC · 2024-07-11

**Soundness:** 2
**Presentation:** 2
**Contribution:** 2
**Rating:** 6
**Confidence:** 3

**Summary:**

The paper introduces a new adapter-based method for continual imitation learning that avoids episodic replay and exhibits better forward and backward transfer and overall performance as compared to prior work. The authors compare their method against baselines on a couple of simulation benchmarks and also provide ablation studies to motivate their design choices.

**Strengths:**

- The paper addresses an important problem of doing continual learning without having to store all the data seen in the past.
- The method is described in detail with experiments provided on a variety of baselines. IsCIL seems to outperform baselines on both simulated benchmarks.
- The authors include ablation studies in the paper to promote their design choices.

**Weaknesses:**

- The paper is a little hard to follow in certain parts. Figure 1 could be made clearer.
- The method assumes access to datasets labeled with sub-goals. This must be added to the limitations.
- Line 140 mentions that a fixed function f is used to encode the observation and goal into a state embedding. Is this fixed function either obtained from the pretraining phase or is a pre-trained encoder of some sort? How does this encoder deal with changes in a non-stationary environment that it might not have been trained on?
- It might be useful to also evaluate IsCiL on the LIBERO benchmark which is developed for such continual learning studies and also provides human-collected demonstrations. This would help highlight the efficacy of the proposed method further.
- Is the choice of using a diffusion base policy for a specific reason?
- The exact lifelong setting for experiments in Table 1 is unclear. From what I understand, the base model is pre-trained on a subset of tasks/objects and new tasks/objects are introduced during the lifelong learning stage. Assuming the table reports multitask performance for all baselines, this raises two questions - (1) Since FWT and BWT are only reported with a single value, does this mean that the training is only done in two stages - pretraining with limited objects and all new objects introduced together? In case the new objects are introduced incrementally, should these numbers be computed for each stage where a new object/task is introduced? (2) How does varying the task order and the initial pretraining set of tasks/objects affect the final performance?
- Lines 233-234 mention that IsCiL exhibits performance ranging between 84.5% and 97.2%. However, in Table 1, I do not see any number as high as 97% and I can see IsCiL performance as low as 68.9% in certain settings. Also, IsCiL does not seem to be “surpassing the oracle baseline for Multi-task learning” as mentioned for any of these cases. Some clarification about how to interpret these results would be helpful.

**Questions:**

It would be great if the authors could address the “Weaknesses” listed above. Also, is there an expanded version of the name of the method - IsCiL?

I am willing to increase my score once these questions have been addressed.

**Limitations:**

The authors have addressed the limitations. I have suggested adding the requirement of sub-goal labeled datasets as a limitation under “Weaknesses”.

---

> ### Author Rebuttal · Authors · 2024-08-06
>
> Thank you for your thorough and insightful review of our paper. Here, we respond to your comments and address the issues.
> > W1. Figure 1 could be made clearer.
>
> We added revised version of Figure 1 in *global rebuttal PDF*.
> > W2. The method assumes access to datasets labeled with sub-goals. This must be added to the limitations.
>
> We will add this limitation in the final version. However, we assert that a sub-goal labeled dataset is common for our setting of imitation learning with multiple long-horizon tasks.
> Here are the reasons for our claim:
> - To evaluate multiple tasks without additional training, the tasks must be distinguishable based on given state information.
> - This assumption is also used in previous baselines such as L2M and TAIL.
> - Particularly for long-horizon tasks, language-based sub-goal labels are commonly used for task label[10].
>
> Moreover, prototype-based skill incremental learning (Section 3.3) can be extended to cases where skills are learned without task information (sub-goals), making it task-agnostic. However, similar to skill-based reinforcement learning, it will require post-training of a high-level policy to learn task information. Addressing this assumption is essential for our future work in lifelong skill-based reinforcement learning.
>
> > W3. Is fixed function(state encoder) f either obtained from the pre-training phase or is a pre-trained encoder of some sort? How does this encoder deal with changes in a non-stationary environment that it might not have been trained on?
>
> We use a very simple concatenation as a fixed encoding function for f. This is defined from the pre-training phase and is some sort of a pre-trained encoder[6]. The reasons for this encoder design choice and its effectiveness in handling non-stationary environments are as follows:
>
> - In adapter-based CiL, task performance depends on the accurate skill retrieval process using the encoded information.
> - The bias of a pre-trained encoder does not guarantee distinction in non-stationary environments. Retraining the encoder could negatively impact overall performance due to distribution shifts, reverting to the continual learning problem.
> - In this retrieval-based system, we can apply various algorithms to improve retrieval accuracy and speed, ensuring that IsCiL remains efficient and accurate despite the stable encoder. For example, IsCiL ensures accurate skill distribution retrieval by utilizing multifaceted bases.
>
> Therefore, IsCiL robustly handles unknown non-stationary environments using multifaceted skill prototypes, even with a fixed encoding function f.
> > W4. It might be useful to also evaluate IsCiL on the LIBERO benchmark.
>
> We demonstrate in the *global rebuttal PDF* that IsCiL can also be applied to LIBERO benchmark.
> > W5. Is the choice of using a diffusion base policy for a specific reason?
>
> Yes, the choice of using a diffusion-based policy is due to its superior performance and ability to handle multi-modal trajectory distributions, as noted in [3,4].
>
> The important point is that IsCiL and other adapter-based baselines are agnostic to the pre-trained model architecture. These inherent advantages of diffusion-based policies create a powerful synergy when combined with the pre-trained model agnostic approach, making IsCiL highly effective in realistic CiL scenarios where the amount of available datasets (expert demonstrations) is limited and tasks and sub-goals can be achieved through multiple paths. Consequently, there was no reason to use a different model architecture for continual imitation learning.
> > W6. The exact lifelong setting for experiments in Table 1 is unclear. (1) the training is only done in two stages? (2) How does varying the task order and the initial pretraining set of tasks/objects affect the final performance?
>
> (1) Our experiment scenario is divided into two parts: Pretraining (stage 0 for convenience) and The CiL scenario(stages 1 to 20). Each CiL stage introduces new tasks, including unseen objects in the pretraining phase. Details of the pretraining tasks and CiL stages are provided in Appendix A.3.
>
> Our metrics are calculated after all CiL stages (20) have been completed. We recorded the success rates of all learned tasks. For each stage (1-20), we calculate FWT, BWT, and AUC. The final reported scores are the averages of all learned tasks(Appendix B.3).
>
> The oracle baseline (multi-task), as described in Section 4, refers to the scenario where all data from each of the 20 stages is stored and used for rehearsal in subsequent stages, allowing the model to be fully trained with this cumulative data. Similarly, we recorded the success rate for each stage's task and reported the numbers similarly to other baselines.
>
> For intuitive understanding, we present the learning curve of Table 1 showing the comprehensive task performance at each stage in the *global rebuttal PDF*.
>
> (2) We added the CiL performance on pre-trained model's quality and task order variations on *global rebuttal Exp 2,3*. The experiments were conducted under the same conditions as Table 1.
> * In Exp 2, the lower the quality of the pre-trained model, the more degradation occurs due to the capability limits of the adapter.
> * In Exp 3, the performance of all tasks at the final stage is not significantly affected. However, the FWT, BWT, and AUC reported in the paper are affected because these scores reflect the interaction of knowledge between tasks throughout the entire scenario, which is influenced by the order.
>
> > W7 Confusing explanation in Lines 233-234.
>
> Thank you for pointing out the ambiguity and the incorrect expression! We intended this sentence to emphasize that our performance achieved a final AUC of 80\% to 97.2\% of the oracle baseline's performance. We will update this in the final version.
> > Q1. Is there an expanded version of the name of the method - IsCiL?
>
> Yes, Incremental skills for Continual Imitation Learning(IsCiL) is the expanded version.

---

> > ### Comment · Reviewer_B1dC · 2024-08-08
> > **Thank you for the rebuttal**
> >
> > I thank the authors for the detailed clarifications and additional experiments. Taking the rebuttal into account, I am raising my score to 6.

---

> ### Author Response · Authors · 2024-08-08
> **Thank you!**
>
> Thank you for raising your score from 4 to 6. We truly appreciate your consideration and constructive feedback. We will incorporate the clarifications and feedback into the final version.
>
> We are particularly pleased that the suggested experiments have allowed us to emphasize the robustness of IsCiL. We are very grateful to the reviewer for proposing these experiments through this discussion.

---

### Official Review · Reviewer_KNda · 2024-07-11

**Soundness:** 3
**Presentation:** 2
**Contribution:** 2
**Rating:** 6
**Confidence:** 4

**Summary:**

Learn a two-layer hierarchy from a sequence of datasets, where the low-level skills are represented by a discrete set of prototypes: vectors that can be mapped to repeated patterns of actions represented by basis functions. The basis function parameters are then passed into a decoder function which takes actions based on the observation and goal. The prototypes are recovered by performing k-means to cluster the data of a particular skill. The frequency of skill selection overall is a score, and the decoder is trained with imitation learning.

**Strengths:**

Introduces a complex but clearly effective system for skill learning.

Shows good results in an important setting of imitation learning from multiple datasets.

Demonstrates unlearning capability, which is useful in some contexts.

**Weaknesses:**

It is not obvious how the skills might be entangled together post-hoc, since the reusability of a skill across tasks might make its unlearning impossible. The experimental results also seem cherry-picked to ensure that this is not an issue, which is probably disingenuous to the actual cause of privacy: whether a component be relearned without any of the information from a particular source.

The experiments appear to be convincing only in the semi and incomplete settings, but it is not entirely clear what the semi or incomplete settings are. Without a clear picture of how these components are defined, it is not clear whether the empirical results actually support the claims made in the introduction.

**Questions:**

Why is Continual Imitation Learning abbreviated CiL? It seems like it should be CIL.

Are there clear ablations on how the many components contribute to the overall performance?

What metric can be used to evaluate the unlearning capability in the context of privacy? Can this be used to ensure particular data is not used? Was this evaluated?

**Limitations:**

see above.

---

> ### Author Rebuttal · Authors · 2024-08-06
>
> Thank you for your thorough and insightful review of our paper. Here, we respond to your comments and address the issues.
> > W1. How the skills might be entangled together post-hoc? since the reusability of a skill across tasks might make its unlearning impossible, the experimental results also seem cherry-picked to ensure that this is not an issue, which is probably disingenuous to the **actual cause of privacy**: whether a component be relearned without any of the information from a particular source.
>
> Here, we provide the detailed process by which skills are connected post-hoc.
>
> * IsCiL incrementally accumulates skills by saving the adapter and prototype pair of each skill for every Continual Imitation Learning (CiL) stage. As described in Section 3.3, once skills are learned and accumulated, the pairs remain immutable after the stage they are learned until an unlearning request is made.
> * During evaluation, the skill retriever search appropriate skill by searching a nearest neighbor of the given state. This retrievable skill accumulation allows the model to improve its performance over time. For example, even if the model initially performs poorly when encountering an unfamiliar state, it can later retrieve relevant skill data if it becomes available. By using the skill prototype, the model can infer the correct action for that state.
>
> We emphasize that the actual cause of privacy in IsCiL is maintained at the **task level**, and this is **not a cherry-picked result for our setting**.
>
> * In Table 3, we demonstrated a task-level unlearning case. In our study, CiL, a task is a sub-goal sequence, and different sub-goal sequences represent different tasks, as described in Section 3.1. Its subsets, overlapping skills, can of course exist through other tasks. Therefore, the unlearning experiment in Section 4.5 aims to make the model behave as if the unlearning task data were never used in training. This process involves deleting the skills (adapter and prototype pairs) learned through the target unlearning task, thereby completely eliminating the impact of the task's data (particular source) had on the model.
>
> * Accordingly, like CLPU, IsCiL meets the actual cause of privacy for target task unlearning: achieving model parameter distribution equality between the unlearned model and the relearned model, which is trained with the same learning algorithm as if the particular source(target unlearning task data) never existed from the start. To ensure this equality, the initialization of the skill adapter was modified to use only the information from the pre-trained model.
>
> * For future directions, unlike our task-level unlearning, completely forgetting the target task while retaining the performance of multiple tasks strongly affected by the skills in the target task will be a very important and challenging area for unlearning.
>
>
> > W2. semi and incomplete settings is not entirely clear. How empirical results actually support the claims made in the introduction?
>
> Semi and Incomplete scenarios refer to CiL situations where comprehensive expert demonstrations are not provided (Section 4.1, Figure 3). We will add a clear explanation of these scenarios in the final version. Concise details are as follows:
>
> * Complete: Consists of 20 CiL stages, each stage incrementally introduces tasks with objects not present in the pre-training stage, along with comprehensive demonstrations.
>
> * Semi: The first 10 stages of the Complete scenario are repeated twice. Each stage includes tasks with incomplete demonstrations, where trajectories for specific sub-goals are missing.
>
> * Incomplete: All stages have the same sequence of tasks as in the Complete scenario, but each stage includes tasks with incomplete demonstrations, where trajectories for specific sub-goals are missing.
>
> For example, in the Semi or Incomplete scenario, if a task composed of sequential sub-goals a-b-c-d is missing sub-goal b, the demonstration will reflect this omission, resulting in a sequence like a-[]-c-d, where [] indicates the missing part. Appendix A.3 provides detailed information about the task sequence and missing parts for each scenario.
>
> We tackle sample efficiency in CiL, which does not require comprehensive demonstrations and rehearsals. Sample efficiency refers not only to the learning efficiency within the stage the sample belongs to but also to the learning and evaluation efficiency across stages. This can be verified through FWT and BWT metrics. Therefore, the high AUC (including FWT and BWT) performance in semi and incomplete scenarios, which require knowledge from other stages to address missing parts, quantitatively validates our sample efficiency.
>
> > Q1. Why is Continual Imitation Learning abbreviated CiL?
>
> We have opted to use 'CiL' for Continual Imitation Learning to avoid confusion, as 'CIL' is commonly used to refer to Class Incremental Learning.
>
> > Q2. Are there clear ablations on how the many components contribute to the overall performance?
>
> The key components of IsCiL are the skill retriever and skill decoder. The performance ablation of the skill retriever is provided in Section 4.7, while the ablation related to the skill decoder is reported in Global Rebuttal Experiments 1 and 2.
>
> > Q3. What metric can be used to evaluate the unlearning capability in the context of privacy? Can this be used to ensure particular data is not used? Was this evaluated?
>
> * In Table 3, we found it meaningless to measure unlearning capability as a metric. This is because ensuring independent adapters for each task resulted in our **unlearned model and relearned model being exactly identical when given the same seed**. Although it is possible to compare the unlearning capability of our policy using the Wasserstein Distance(WD) between the output distributions of the unlearned model and the relearned model, following [2], this comparison also becomes meaningless for the same reason.

---

> > ### Comment · Reviewer_KNda · 2024-08-09
> > **Response to Authors**
> >
> > I appreciate the clarifications and believe that the additions will strengthen the paper. I am happy to raise my score.

---

> > > ### Author Response · Authors · 2024-08-09
> > > **Thank you!**
> > >
> > > Thank you for raising your score from 5 to 6. We truly appreciate your consideration and constructive feedback. We will, of course, incorporate this discussion into the final version.
> > >
> > > Additionally, although IsCiL primarily focuses on task unlearning, researching how to maintain CiL performance while ensuring privacy at the skill level is a challenging but promising area for future work. The question of which metrics to use for measuring unlearning privacy was particularly insightful and will be invaluable for advancing this skill unlearning approach.

---

### Author Rebuttal · Authors · 2024-08-06

We sincerely thank all reviewers for their thoughtful reviews and greatly appreciate the insightful feedback on our work. In this section, we include experiments (with PDF) and references to address the comments provided.

## **Experiment**
---
> Exp 1. Skill adapter rank ablation. [KNda Q2 | kNto W3]

|||Evolving|Kitchen|-complete|Evolving|World|-complete|
|-|-|-|-|-|-|-|-|
|Rank|Baselines|FWT|BWT|AUC|FWT|BWT|AUC|
|1|L2M-g|30.16|2.59|32.96|56.83|-16.93|41.60|
|1|TAIL-g|93.21|-54.30|45.68|76.95|-47.93|48.62|
|1|IsCiL|89.18|2.73|91.57|73.63|-3.31|70.91|
|4|L2M-g|38.19|-6.50|32.33|64.19|-19.34|48.62|
|4|TAIL-g|85.28|-49.90|41.54|90.02|-56.76|39.53|
|4|IsCiL|79.31|11.03|89.76|81.69|2.70|84.30|

We conduct an ablation study on the performance of Continual Imitation Learning (CiL) based on the rank of the skill adapter. Overall, the 1-rank adapter in Evolving Kitchen shows sufficient or even superior adaptation performance. In contrast, in Evolving World, the 1-rank adapter results in lower overall performance, indicating that some skills cannot be fully learned with a 1-rank adapter, leading to a decline in performance.

> Exp 2. Skill decoder pre-trained model quality ablation. [KNda Q2 | B1dC W6]

|||Evolving|Kitchen|-complete|Evolving|Kitchen|-incomplete|
|-|-|-|-|-|-|-|-|
|Baselines|Pre-training|FWT|BWT|AUC|FWT|BWT|AUC|
|TAIL-$\tau$|1obj|72.77|0.00|72.77|28.75|0.00|28.75|
|-|2obj|87.24|0.00|87.24|35.86|0.00|35.86|
|-|4obj|86.24|0.00|86.24|33.76|0.00|33.76|
|IsCiL|1obj|60.01|2.07|62.13|42.08|5.39|46.97|
|-|2obj|78.88|6.42|84.92|56.67|11.95|67.29|
|-|4obj|79.31|11.03|89.76|61.81|13.71|74.04|

We conduct an ablation study on the performance changes based on the quality of the pre-trained model (skill decoder). The quality of the pre-trained model varies with the number of objects included in the tasks used to pre-train the model. A decrease in the quality of the pre-trained model leads to a performance drop in both TAIL-$\tau$ and IsCiL (from 4 objects to 1 object).


> Exp 3. CiL scenario task sequence variation analysis. [B1dC W6]

|4 scenario|FWT|BWT|AUC|
|-|-|-|-|
|TAIL-$\tau$|86.24|0.00|86.24|
|IsCiL|78.19|7.42|86.05|

We report the average performance for **four different task sequences** in **Evolving Kitchen-complete**. The performance of all tasks at the final stage is not significantly affected. Since TAIL-$\tau$ learns independently for each task ID, there was no performance change with different sequences, and IsCiL also showed similar performance.

## **PDF contents**
---
1. LIBERO Experiment [B1dC W5 | Dhgs Q1]
2. Revised Figure 1 [B1dC W1]
3. Learning curve of Table 1 [Dhgs Q1 | B1dC W6(1)]

## **References**
---
[1] Liu, Bo, Qiang Liu, and Peter Stone. "Continual learning and private unlearning." Conference on Lifelong Learning Agents. PMLR, 2022.

[2] Tarun, Ayush Kumar, et al. "Deep regression unlearning." International Conference on Machine Learning. PMLR, 2023.

[3] Pearce, Tim, et al. "Imitating Human Behaviour with Diffusion Models". International Conference on Learning Representations, 2023.

[4] Wang, Zhendong, et al. "Diffusion Policies as an Expressive Policy Class for Offline Reinforcement Learning". International Conference on Learning Representations, 2023.

[5] Hu, Edward J., et al. "LoRA: Low-Rank Adaptation of Large Language Models". International Conference on Learning Representations, 2022.

[6] Schmied, Thomas, et al. "Learning to Modulate pre-trained Models in RL." Advances in Neural Information Processing Systems 36. 2024.

[7] Douze, Matthijs, et al. "The Faiss Library". arXiv, 2024.

[8] Wan, Weikang, et al. Lotus: Continual Imitation Learning for Robot Manipulation through Unsupervised Skill Discovery. 2024.

[9] Bruce, Jake, et al. ‘Learning About Progress From Experts’. International Conference on Learning Representations, 2023.

[10] Shridhar, Mohit, et al. "Alfred: A benchmark for interpreting grounded instructions for everyday tasks." Proceedings of the IEEE/CVF conference on computer vision and pattern recognition. 2020.

---

### Decision · Program_Chairs · 2024-09-25

**Decision:**

Accept (poster)

**Comment:**

This submission introduces a method for continual imitation learning by prototype-based skill incremental learning. Low-level skills are represented by discrete prototypes (like basis functions), and their parameters, together with observations and goals, are passed into a decoder for action execution. The main contribution is that the proposed method eliminates the need for episodic replay, offering improved forward and backward transfer, and outperforms prior related works. The authors evaluated the method on Franka-Kitchen and Meta-World, and the method is shown to have strong adaptability to new tasks.

All reviewers agreed that the proposed method is interesting both method-wise and application-wise. The performance gain is strong as well. Thus, the meta reviewer recommends "accept" for this submission.